Methods

# FrozONE: quick cell nucleus enrichment for comprehensive proteomics analysis of frozen tissues

Lukas A Huschet[1],*, Fabian P Kliem[1],*, Peter Wienand[2], Claudia M Wunderlich[2], Andrea Ribeiro[3], Isabel Bustos-Martínez[4], Ángel Barco[4], F Thomas Wunderlich[2], Maciej Lech[3] , Maria S Robles[1]

Subcellular fractionation allows for the investigation of compartmentalized processes in individual cellular organelles. Nuclear enrichment methods commonly employ the use of density gradients combined with ultracentrifugation for freshly isolated tissues. Although it is broadly used in combination with proteomics, this approach poses several challenges when it comes to scalability and applicability for frozen material. To overcome these limitations, we developed FrozONE (Frozen Organ Nucleus Enrichment), a nucleus enrichment and proteomics workflow for frozen tissues. By extensively benchmarking our workflow against alternative methods, we showed that FrozONE is a faster, simpler, and more scalable alternative to conventional ultracentrifugation methods. FrozONE allowed for the study, profiling, and classification of nuclear proteomes in different tissues with complex cellular heterogeneity, ensuring optimal nucleus enrichment from different cell types and quantitative resolution for low abundant proteins. In addition to its performance in healthy mouse tissues, FrozONE proved to be very efficient for the characterization of liver nuclear proteome alterations in a pathological condition, diet-induced nonalcoholic steatohepatitis.

## Introduction

Cellular compartmentalization has played a pivotal role in the evolution of life and its complexity and has allowed the development of specialized regions within cells, known as organelles, each dedicated to specific tasks (Honigmann & Pralle, 2016). Understanding molecular processes specifically occurring in these cellular niches requires efficient subcellular fractionation or isolation techniques to yield compartmentalized organelles for their individual investigation. Quantitative proteomics is a very powerful technology that allows the characterization of tissue and cellular processes with great depth. When combined with organelle enrichment methods, it also permits the study of subcellular protein localization, which is crucial for understanding protein function, as well as the cellular processes and pathologies linked to their spatial misplacement (Christopher et al, 2021). In addition, subcellular proteomics enable deeper coverage of spatially confined, low abundant proteins that are otherwise difficult to detect because of the high dynamic range in whole tissue or cell protein extracts (Wu & Han, 2006).

The cell nucleus is an organelle that stores the genetic information necessary to orchestrate all cellular processes (Rippe, 2007). The development of nucleus enrichment methods dates back to the 1940s (Claude, 1943; Claude & Potter, 1943), commonly employing the use of density gradients combined with ultracentrifugation to separate cellular compartments. Among them, sucrose-based gradient methods using freshly isolated tissues have become the gold standard, as sucrose is inexpensive and water-soluble and allows collection of nuclei with a high degree of purity (Lee et al, 2010). However, this approach poses several limitations to the scalability and robustness of proteomics workflows: (i) ultracentrifuges generally have a small number of slots, limiting parallel processing required in large-scale studies; (ii) not every laboratory has access to an ultracentrifuge; (iii) preparation of individual gradients and posterior downstream processing is time- and resource-consuming; and (iv) the necessity of using fresh tissue hinders the exploration potential of stored samples such as frozen human specimens or biopsies.

To address these issues, we established FrozONE (Frozen Organ Nucleus Enrichment), a subcellular proteomics workflow that is reproducible, robust, quick, and scalable, for the preparation and analysis of nuclear proteomes from frozen tissues without the need for ultracentrifugation. We extensively benchmarked our workflow and found that FrozONE competed with conventional gradient

---

[1]Institute of Medical Psychology and Biomedical Center (BMC), Faculty of Medicine, LMU, Munich, Germany   [2]Max Planck Institute for Metabolism Research, Center for Molecular Medicine Cologne (CMMC) and Policlinic for Endocrinology, Diabetes, and Preventive Medicine (PEDP), University Hospital Cologne, Cologne, Germany   [3]LMU Klinikum, Medizinische Klinik und Poliklinik IV, Ludwig-Maximilians-Universität München, Munich, Germany   [4]Instituto de Neurociencias (Universidad Miguel Hernández—Consejo Superior de Investigaciones Científicas), Alicante, Spain

Correspondence: crobles@med.lmu.de
Andrea Ribeiro's present address is School of Medicine, Klinikum rechts Der Isar, Department of Nephrology, Technical University of Munich, Munich, Germany.
*Lukas A Huschet and Fabian P Kliem contributed equally to this work

centrifugation methods using both frozen and fresh tissues, in terms of quantitative depth and reproducibility of nuclear proteins. Moreover, we demonstrated its performance with three distinct organs, brain, liver, and kidney, and its availability to resolve their cellular and spatial proteome heterogeneity. Finally, as a proof of principle, we employed FrozONE to characterize alterations of the liver nuclear proteome caused by a high-fat diet (HFD), conventionally used to induce nonalcoholic steatohepatitis (NASH). Our data revealed that a NASH-inducing diet rewires the nuclear abundance of numerous proteins in the mouse liver, including a subset of transcription factors that has been associated with metabolic disruption. This finding highlights FrozONE's potential for use in clinical research.

## Results

### Comparative proteomics analysis of FrozONE and common nucleus enrichment methods

To provide a method that allows for the comprehensive characterization of nuclear proteomes from frozen tissues in a rapid, robust, and scalable manner, we developed FrozONE. This workflow combines the use of a nucleus enrichment kit that does not require ultracentrifugation, commonly used to obtain nuclear preparations for RNAseq (Habib et al, 2017; Bravo González-Blas et al, 2020; Sun et al, 2020), with mass spectrometry (MS)–based quantitative proteomics. Using three different mouse tissues (brain, liver, and kidney), we benchmarked the performance of FrozONE, comparing it with two density gradient methods using ultracentrifugation of frozen tissues (iodixanol frozen and sucrose frozen) (Kim et al, 2015; Ragazzini et al, 2019; Strzelecki et al, 2022), and the traditional gold standard method that uses fresh material and sucrose-based gradient ultracentrifugation (sucrose fresh) (Kim et al, 2015; Ragazzini et al, 2019; Strzelecki et al, 2022) (Fig 1A). We enriched nuclei from all tissues (n = 3) using all four protocols and measured the proteomes by single-shot MS-based quantitative proteomics with data-independent acquisition (DIA) (see the Materials and Methods section).

Although the methods yielded varying numbers of total proteins (all quantified confidently between replicates), the fraction of proteins annotated as nuclear (see the Materials and Methods section) was quite similar between methods for each individual tissue (from 2,500 to almost 4,000 corresponding to 45–49% of the total quantified proteins), indicating that none of the methods systematically outperform the others (Fig 1B). Overall, the fraction of proteins that were robustly quantified in every replicate with FrozONE was 76% (average over tissues) for all proteins and 81% for proteins annotated as nuclear. This fraction is very comparable to the values obtained with the other methods (Fig S1A). We therefore conclude that the use of our quick and low-speed centrifugation workflow with frozen tissues, FrozONE, achieved equivalent reproducible quantification depth of nuclear proteins as classical, more tedious gradient ultracentrifugation–based methods.

After quantification performance, we next assessed the reproducibility of proteomes across methods and tissues by principal

component analysis (PCA). The separation of proteomes in the two main components showed FrozONE as the method with the highest correlation of intensities between biological replicates for all tissues (average Pearson's r > 0.95, Figs 1C and S1B for nuclear and non–nuclear-annotated proteins). FrozONE proteomes showed the least separation in all tissues, even less than those from the gold standard, sucrose fresh, whereas brain from sucrose frozen and kidney and liver from iodixanol scattered far apart. Overall, we observed that the coefficient of variation (CV) of FrozONE protein quantifications was consistently among the lowest across methods for total, nuclear, and non–nuclear-annotated proteins, highlighting the high quantitative reproducibility of our workflow. Furthermore, CVs from nucleus proteomes were generally lower than those from whole-cell proteomes (Fig S1C), indicating a high quantitative reproducibility of nucleus proteome preparations from frozen tissues. Similar quantification reproducibility and robustness were observed for nuclear subgroups of proteins, such as those from nuclear membrane, nucleoplasm, nucleolus, and chromatin binding annotations (GOCC), as expected better than the CVs from the same annotations in whole-cell preparations (Fig S1A and C). FrozONE and the classical sucrose-based gradient ultracentrifugation with fresh tissue produced the most reproducible and robust proteomes with a high degree of overlap (73–87%) and a similar total number of nuclear or non–nuclear-annotated proteins (Fig S2A). Therefore, all further comparisons focused on these two methods. Both produced equivalent dynamic ranges of protein abundance in all tissues, with histones being among the most intense proteins, as expected from efficiently enriched nucleus fractions (Fig 1D). When compared to whole-cell preparations, FrozONE enriched nuclear proteins as efficiently as the gold standard (Fig S2B), showcasing FrozONE's competitiveness to comprehensively characterize nuclear proteomes.

As expected, enrichment analysis of significantly up-regulated proteins in FrozONE nuclei versus whole-cell lysate (WCL) proteomes showed an overrepresentation of nuclear-related terms (i.e., nucleoplasm, nucleolus, and nucleus) and an underrepresentation of non-nuclear terms (i.e., mitochondria, peroxisome, Golgi apparatus, and cytoplasm) in all tissues (Table S1). Indeed, the total number of proteins annotated as localizing to these non-nuclear organelles is not substantial and similar when comparing FrozONE and sucrose fresh (Fig S3A). To further show the enrichment capacity of FrozONE compared to the gold standard method, we compared protein intensities in nuclear-enriched fractions versus WCL. Overall, we found that FrozONE enriched nuclear proteins equal or better, depending on the tissue, than sucrose fresh (Fig S3B). In contrast, proteins annotated from other cellular organelles show clear de-enrichment in FrozONE nuclear preparations, again to a better or similar degree as sucrose fresh preparations (Fig S3C). We can thus conclude that no organelle is prone to substantially "contaminate" FrozONE preparations to a higher degree than in current standard nucleus enrichment methods.

We then investigated the capability of both methods to quantify low abundant proteins from a wide variety of nuclear processes (Cleaver, 2004; Brickner et al, 2019; dos Santos & Toseland, 2021; Stewart, 2022), such as transcription factors (TFs), chromatin modifiers, DNA repair machinery, and nuclear transport machinery.

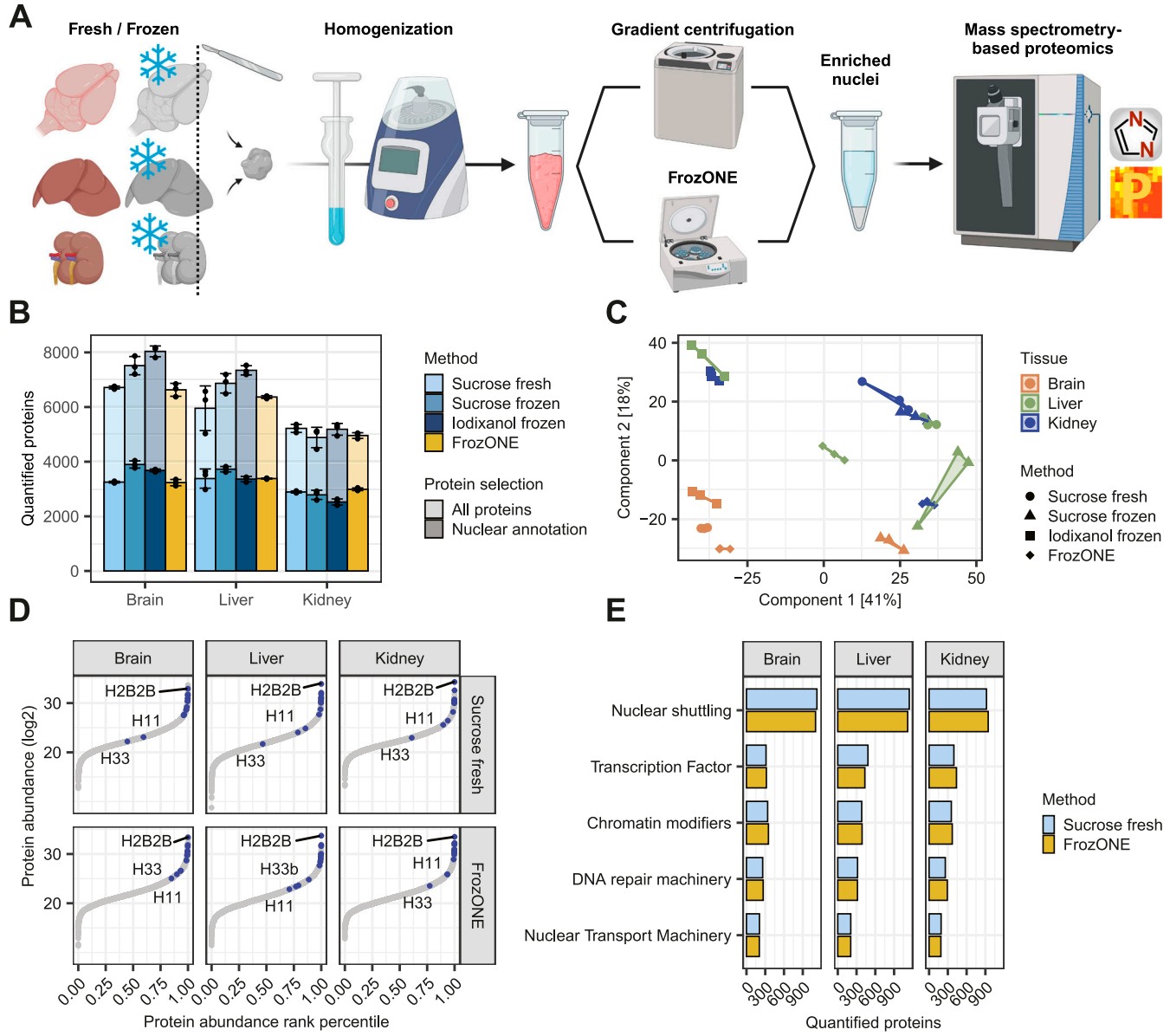

**Figure 1. Comparative proteomics analysis of FrozONE and common nucleus enrichment methods in mouse tissues.**
**(A)** Workflow of nucleus enrichment methods (see the Materials and Methods section for detail). **(B)** Barplot of the mean number of proteins quantified across at least two biological replicates in each tissue and method (n = 3). Light-colored bars denote total protein quantifications, whereas dark bars denote nuclear-annotated proteins (see the Materials and Methods section). Error bars indicate the SD. **(C)** Principal component analysis of protein intensities in biological replicates (n = 3) from each organ and nucleus enrichment method. **(D)** Scatter plot of protein abundance sorted by abundance rank percentile comparing FrozONE and sucrose ultracentrifugation with fresh tissue. Common histones between methods are labeled in blue. **(E)** Barplot of the mean number of proteins quantified across biological replicates (n = 3) in FrozONE and sucrose gradient with fresh tissue for important groups of nuclear proteins.

FrozONE quantified, independently of the tissue, hundreds of these typically low abundant nuclear process proteins, such as more than 400 TFs and chromatin modifiers. These numbers were equal, or in some cases higher, than those obtained with the gold standard method (Fig 1E). Collectively, these benchmark data show that our newly established proteomics workflow FrozONE is a strong competitor to conventional and tedious nuclear enrichment methods providing robust and reproducible protein quantifications, highlighting its potential for large-scale analysis of nuclear proteomes using frozen material.

## FrozONE allows resolution of tissue-specific nuclear proteomes

Having demonstrated that FrozONE performs as well as the current gold standards for nucleus enrichment from not only fresh but also frozen tissues, we sought to investigate its potential to resolve tissue-specific nuclear protein signatures. To this end, we analyzed and compared FrozONE proteomes obtained from mouse brain, liver, and kidney. From the total 8,928 quantified proteins (Table S2), we found 4,101 common to all three tissues, and 5–20% of each proteome exclusive to their tissue (Fig 2A). Tissue-specific protein

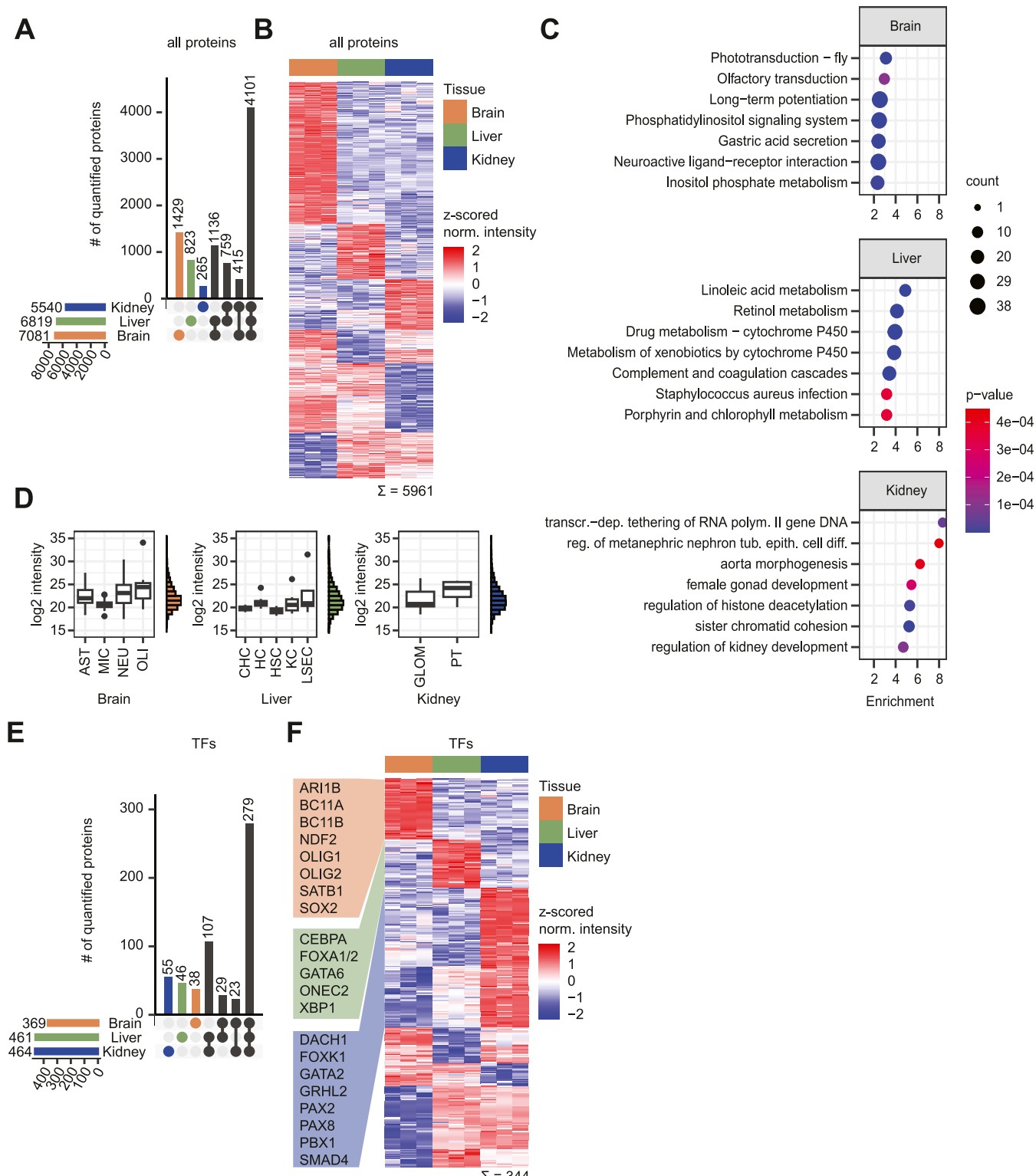

**Figure 2. FrozONE allows resolution of mouse tissue–specific nuclear proteomes.**
**(A)** Upset plot illustrating the distribution of protein quantifications between tissues of 8,928 proteins quantified in at least 2/3 replicates in at least one tissue.
**(B)** Heatmap showing the z-scored label-free quantification of log₂-transformed intensities of 5,961 proteins significant in an ANOVA (FDR = 0.001). **(A, B, C)** Top seven enrichment analysis results of tissue-up-regulated (B) and tissue-exclusive proteins (A). For brain and liver, KEGG terms, and for kidney, GOBP terms are shown.
**(D)** Boxplots showing average log₂ label-free quantification intensities of nuclear cell-type marker proteins for each tissue and histograms showing overall intensity distributions for each tissue. AST, astrocyte; MIC, microglia; Neu, neuron; OLI, oligodendrocyte; CHC, cholangiocyte; HC, hepatocyte; HSC, hepatic stellate cell; KC, Kupffer

signatures can be determined not only by exclusive presence but also by differential abundance of proteins (Wang et al, 2019). We therefore compared protein abundance in all tissues (see the Materials and Methods section) and found that 5,961 proteins (67%) showed statistically significant differences in their abundance across tissues (one-way ANOVA, S0 = 0, FDR < 0.05). Furthermore, 13–37% of these proteins exhibited exclusive up-regulation in only one of the respective three tissues (Fig 2B). By combining tissue-specific up-regulated and tissue-specific exclusively detected proteins, we defined lists of tissue signature proteins (Table S2). Enrichment analysis (Fisher's exact test, FDR < 0.02) of these signature proteins highlighted, as expected, enriched processes with tissue-related functions, such as sensory processing and long-term potentiation in brain, lipid and drug metabolism in liver, and nephron cell differentiation and development in kidney (Fig 2C). Annotations of biological functions that seem specific to a different tissue are generally due to the alternative roles of those annotation-containing proteins in other tissues. For example, proteins from gastric acid secretion are involved in calcium signaling, a key process for brain function. Taken together, our analysis demonstrates the capability of FrozONE to characterize tissue-specific nuclear protein signatures.

Given the high degree of cellular heterogeneity within organs, we next aimed at confirming the capability of FrozONE to enrich nuclei from different cell types within organs. To this end, we interrogated our data for cell-type markers defined by published whole-cell proteomics studies, considering only proteins annotated as nuclear (see the Materials and Methods section): 192 markers from 4 brain cell types (Sharma et al, 2015), 54 markers from 5 liver cell types (Azimifar et al, 2014), and 20 markers from 2 kidney cell types (Sigdel et al, 2020). From all of them, we detected 71% (145) of brain, 40% (23) of liver, and 65% (13) of kidney cell-type markers that overall represent all the 11 cell types (Fig 2D). Taking into consideration the fact that Sharma et al (2015) and Azimifar et al (2014) used peptide fractionation to enhance quantification depth, while we relied on DIA, the coverage of our single-shot FrozONE nuclear proteomes is remarkable. In addition to its great depth, our workflow's high quantification reproducibility and robustness allowed us to recapitulate reported distributions of cell types in each tissue by comparing the intensities of nuclear-annotated cell-type marker proteins within the tissues (Fig 2D). For example, intensities of kidney proximal tubular cell marker proteins were higher than those of glomerular cell markers, in line with their respective 66% (Balzer et al, 2022) and 5% (Appel & Radhakrishnan, 2012) proportion in this tissue. Similarly, intensity differences of brain cell-type markers recapitulated the distributions of the corresponding cell populations: 10% microglia (Ochocka & Kaminska, 2021), 10–20% astrocytes (Sun et al, 2017), 20% oligodendrocytes, and 50% neurons (Valério-Gomes et al, 2018). Notably, the proportion of neurons and oligodendrocytes were even better reflected when using individual conventional markers (Fig S4A). The same was also observed for the distribution of the different liver cell types: 3–5% cholangiocytes (MacParland et al, 2018), 5–10% hepatic stellate cells (Kamm &

McCommis, 2022), 15% Kupffer cells (Sitia et al, 2011), 15–20% liver sinusoidal endothelial cells (LSECs) (Du & Wang, 2022), and 80% hepatocytes (Blouin et al, 1977; Bogdanos et al, 2013). Only oligodendrocytes and LSECs were not precisely recapitulated because of the large variability of marker intensities. We conclude that FrozONE is a robust method to profile nuclear proteomes from highly heterogeneous organs and can sufficiently quantify markers from low abundant cell types.

Besides general tissue and cell type–specific characterization, we were particularly interested in exploring potential differences in protein abundance of TFs across tissues, as TFs drive gene expression programs to control development and function in an organ- and a cell type–specific manner (Hammonds et al, 2013; Jiménez et al, 2023). When filtering our quantified proteins for TFs listed in the Animal Transcription Factor Database 4.0 (TFDB 4.0 [Shen et al, 2023]), we identified around 10% to be exclusive in one of the tissues (Fig 2E). When comparing the abundance of TFs commonly present in all tissues, we found 344 of them showing statistically significant differences across tissues (one-way ANOVA, S0 = 0, FDR < 0.05), of which 10–20% were exclusively up-regulated in single tissues (Fig 2F). Combining both tissue-exclusive and tissue-up-regulated TFs, we defined TF signature datasets in brain (65), liver (73), and kidney (145) (Table S2). These signatures included TFs with known key roles in the development of the respective tissues (Fig 2F), such as the neurodevelopmental factors BC11, NDF2, and SOX2 (Lennon et al, 2017; Simon et al, 2020; Tutukova et al, 2021; Mercurio et al, 2022). Likewise, tissue signatures also contained TFs that play cell type–specific roles, such as ARI1B and SATB1, which regulate neurogenesis and senescence, respectively (Cancio-Bello & Saez-Atienzar, 2020; Moffat et al, 2021), and OLIG1/2, which is required for oligodendrocyte differentiation (Dai et al, 2015; Tsigelny et al, 2016).

Together, our data indicate that FrozONE is a suitable method to study, profile, and classify nuclear proteomes in different tissues with complex cellular heterogeneity. Our method ensures enrichment of nuclei from diverse and low abundant cell types, thus providing quantitative resolution of low abundant proteins.

### Spatial resolution of nuclear proteomes from distinct mouse brain areas

The brain is a heterogeneous tissue with a high degree of diversity in regard to anatomical structure, cellular composition, and function (Miterko et al, 2018). Robust and scalable methods able to generate comprehensive nuclear proteomes from distinct brain areas could provide relevant molecular insights into the mechanisms that drive context-specific spatial gene expression (Kang et al, 2011) and epigenetic reprogramming in healthy and pathological brains (Isles, 2018).

To test whether FrozONE is suitable to characterize brain area specific nuclear proteomes, we applied it to two functionally distinct brain regions, the hippocampus (HC), involved in memory and emotion (Lisman et al, 2017), and the hypothalamus (HT), involved in

---

cell; LSEC, liver sinusoidal endothelial cell; GLOM, glomerulus; PT, proximal duct. **(E)** as A but filtered for transcription factors (TFs). **(F)** as B but filtered for TF. Marker TFs relevant for their tissue transcriptional machinery are indicated.

hormonal and homeostatic balance, and the regulation of physiological daily rhythms such as the sleep–wake cycle and food intake (Lechan & Toni, 2000; Van Drunen & Eckel-Mahan, 2021).

FrozONE-enriched nuclear fractions obtained from HC and HT yielded more than 2,000 quantified nuclear-annotated proteins, and similar ratios of nuclear annotated over total quantified proteins as full forebrain preparations (49% and 54%, respectively, compared with 48% in forebrain; Fig 3A). These ratios exceeded, or were equal to, those reported in recent studies using nuclear enrichment strategies by proximity labeling or differential centrifugation of either total or specific brain areas (Dumrongprechachan et al, 2021; Herbst et al, 2021; Kandigian et al, 2022) (Fig S5A). As those studies employed diverse methodology, we opted for generating reference nuclear proteomes, using the same MS methods, from high-purity nuclear preparations. To do so, we used fluorescence-activated nucleus sorting (FANS) and a mouse model that conditionally expresses SUN1-GFP in CaMKIIα-positive hippocampal neurons (Fernandez-Albert et al, 2019) (see the Materials and Methods section). Proteomic measurements of FANS nuclei yielded 3,864 quantified proteins, from which 57% (similar to FrozONE brain and HC samples) were annotated as nuclear (Table S3), despite being highly pure nuclei preparations. This points toward an incompleteness of the protein annotation database, thus concluding that FrozONE nucleus enrichment is quite efficient and comparable to, in principle, purer nuclei preparations. Overall, almost 80% of proteins quantified in SUN1-GFP neuronal hippocampal nuclei were also quantified in the FrozONE hippocampus preparations. Those proteins exclusively quantified in FrozONE hippocampus samples were enriched in non-neuronal cell-type markers, as expected from these samples that contained all cell-type diversity in the hippocampal region (Fig S5B and Table S3). Despite not being contained in FANS proteomes, almost 50% of these FrozONE-exclusive proteins have nuclear annotations, like the overall percentage seen in the rest of FrozONE, sucrose fresh, and even FANS nucleus-exclusive or total fractions (Figs S3A and S5B). Dimensionality reduction in the three FrozONE brain nuclear proteomes using PCA revealed high reproducibility within triplicates and a clear separation between brain regions, indicating distinct nuclear proteome composition (Fig 3B). This separation is based on the differential abundance of commonly quantified proteins across samples (a requirement for PCA). Hence, it is not surprising that most HC or HT proteins were also found in the total brain (Fig S5C).

The coverage of specific functional nuclear protein groups was also very similar in total forebrain, HC, and HT, relative to the number of total quantified proteins (Fig 3C), differing by less than 3% between areas. As observed, when we compared different organs (Fig 1D), protein abundance ranges of HC and HT closely resembled those of the forebrain, with histones being distributed toward the highest abundance percentiles (Fig 3D).

As our FrozONE workflow exhibited great separation between brain region–specific proteomes, we next focused on whether we could determine region-specific TFs in HC and HT, which play a key role in brain and cellular lineage development (Scotting & Rex, 1996; Thiel, 2006) and often show area-specific expression and function (Mason et al, 2009; Co et al, 2020). Similar to total proteomes, most of the TFs quantified in the nuclear preparations of HT and HC (217 and 245, respectively) were also found in total forebrain.

However, FrozONE allowed us to resolve a few TFs in each area that were not detected in the others (Fig S5D). Overall, our data show the power of FrozONE to quantify low abundant nuclear functional proteins in distinct brain areas, quantifying overall more TFs than previous studies that performed comprehensive whole tissue proteomics with elaborate fractionation strategies to boost their quantification depth (Sharma et al, 2015; Liu et al, 2023) (Fig S5E).

In addition to TFs exclusively detected in brain areas, several other TFs (4 in HT and 27 in HC) showed statistically significant abundance differences (one-way ANOVA, S0 = 0, FDR < 0.05, Fig S5F). Considering both exclusive and significantly up-regulated TFs, we defined brain area–specific TF signatures, of which more than 25% are reported to be essential for neuronal development (Fig 3E, Table S3). For example, NFIB is identified as a HC and PITX2 as a HT marker, both of which are known to regulate the development of the respective brain area (Mason et al, 2009; Waite et al, 2013).

Our brain area–derived FrozONE proteomes also contained a large fraction of proteins annotated with functions in neurodegenerative diseases (183 detected out of 288 present in the KEGG pathway database) such as amyotrophic lateral sclerosis (ALS), and Alzheimer's, Huntington's, or Parkinson's disease (Fig 3H). As aberrant nuclear mechanisms, including nucleocytoplasmic transport, chromosomal instability, nuclear inclusions, and RNA processing and transcription, are often a hallmark of neurodegenerative diseases, precise profiling of nuclear protein signatures in frozen specimens of patients affected by neurodegeneration could have a great potential for the molecular understanding of these chronic conditions (Nelson et al, 2022).

In summary, FrozONE is proficient at discerning nuclear proteome patterns specific to small brain regions such as the HC and HT. Consequently, it offers a great potential for the investigation of brain area–specific functions in healthy states and pathological conditions.

### High-fat diet rewires the nuclear proteome of mouse liver

After demonstrating that FrozONE allows deep nuclear proteome quantification from frozen organs and can resolve tissue-specific protein signatures, we next explored its applicability to the identification of alterations in the nuclear compartment driven by pathological conditions. We chose diet-induced nonalcoholic steatohepatitis (NASH), a condition with increased prevalence in developed countries that has been linked to hepatocarcinoma (Pouwels et al, 2022; Teng et al, 2023). Development and progression of NASH involves transcriptional changes related to lipid and glucose homeostasis, as well as other associated processes such as inflammation and fibrosis (Steensels et al, 2020). We therefore employed FrozONE to investigate changes in the nuclear proteome composition of livers from mice fed a NASH-inducing high-fat diet (HFD) (see the Materials and Methods section), compared with mice fed a control diet (CD).

We found that overall protein quantification, percentage of nuclear protein annotations (55% and 53% in CD or HFD, respectively), and specific nuclear protein groups were similar in nuclear preparations of livers from both diets (Fig S6A and B). We therefore conclude that tissue composition, in this case, higher lipid content in HFD livers, does not affect FrozONE's performance. Next, we

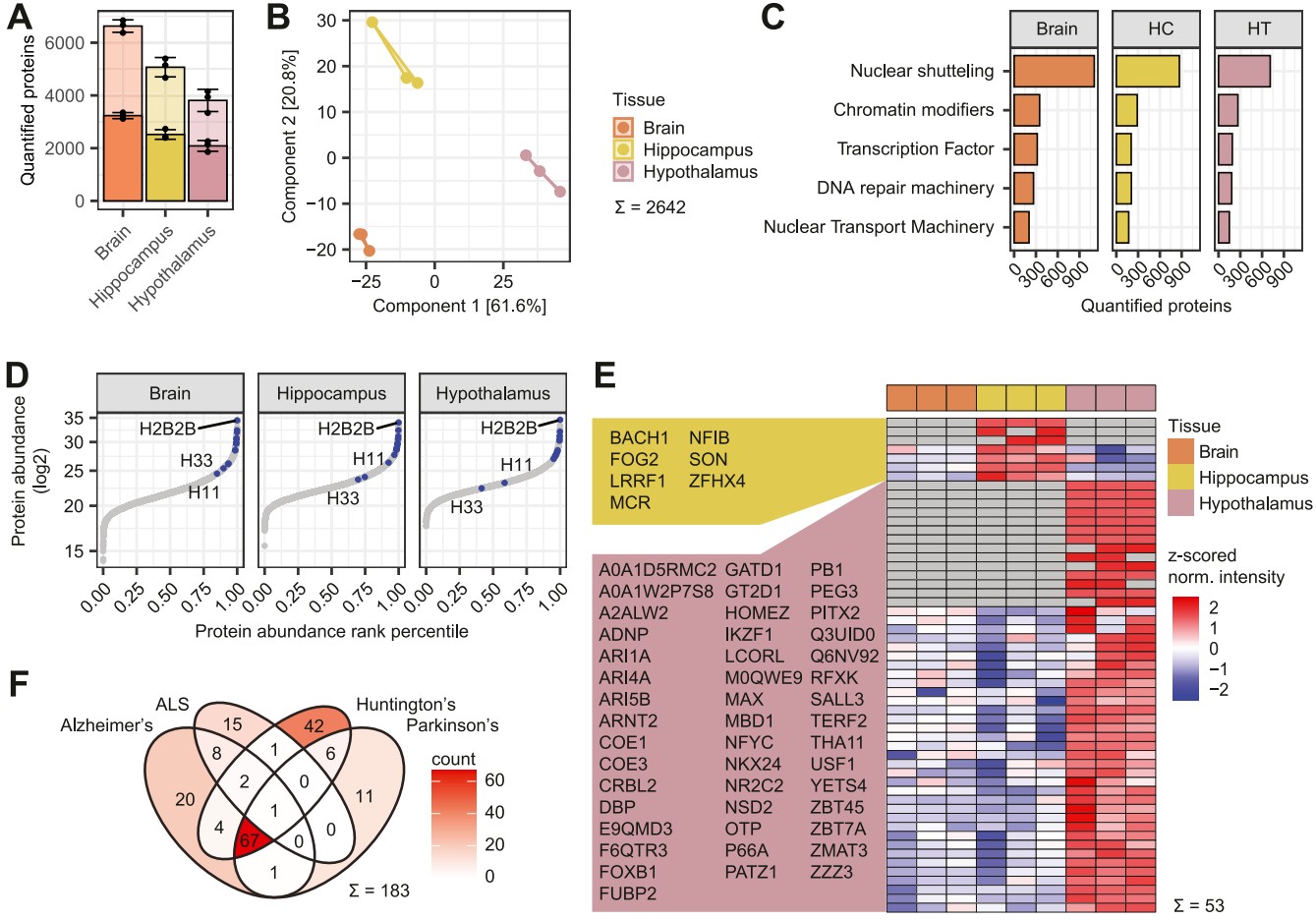

**Figure 3. Spatial resolution of nuclear proteomes from distinct mouse brain areas.**
**(A)** Barplot with the mean number of proteins quantified with FrozONE across three biological replicates from brain areas hippocampus (HC), hypothalamus (HT), and entire forebrain (Brain). Light-colored bars denote total protein quantifications, whereas dark bars denote nuclear-annotated proteins (see the Materials and Methods section). Error bars indicate the SD. **(B)** Principal component analysis of FrozONE proteomes from each brain area in three biological replicates. **(C)** Barplot of the mean numbers of proteins quantified across three biological replicates in the brain areas for important groups of nuclear proteins. **(D)** Scatter plot of protein abundance sorted by abundance rank percentile comparing the brain areas. Common histones between areas are labeled in blue. **(E)** Heatmap showing the z-scored normalized $\log_2$-transformed intensities of 53 HC and HT marker transcription factors (ANOVA-significant S0 = 0, FDR = 0.05 or exclusive). **(F)** Venn diagram showing the number of proteins quantified in the areas with KEGG pathway annotation for selected neurodevelopmental diseases.

compared protein abundance in the nuclear-enriched liver preparations from the two diets. We found that 26% and 3.2% (1,755 and 215 out of the total 6,721) of quantified proteins showed statistically significant abundance differences (*P*- and *Q*-value < 0.05, $\log_2$ fold change > 1, Fig 4A, Table S4) in HFD and CD livers, respectively. Our data thus show that diet-induced metabolic challenge rewires the nuclear proteome composition in the liver, which is consistent with the extensive transcriptional changes reported in this organ under the same metabolic condition (Dorn et al, 2014; Steensels et al, 2020).

Furthermore, we observed that nuclear-annotated proteins with statistically significant higher levels under HFD are enriched in metabolic-related pathways, such as lipid and fatty acid metabolism, NADP- and NAD-related processes, and lipid binding (Fisher's exact test, FDR < 0.05, Table S4). In particular, we detected HFD-driven higher nuclear accumulation of several enzymes involved in acetyl-CoA metabolism. As acetyl-CoA is used as an acetyl

group donor for histone acetylation, our data indicate that higher nuclear levels of those enzymes could be a molecular underpinning driving reported HFD-associated effects in histone acetylation levels (Arias-Alvarado et al, 2021). For example, higher levels of ATP citrate lyase (ACLY), a key enzyme mediating the conversion of citrate to acetyl-CoA in the nucleus, in response to HFD could lead to an increase of nuclear acetyl-CoA levels to be subsequently stored in the form of histone acetylation (Boon et al, 2020), linking thus metabolic to epigenetic state. In addition to epigenetic alterations, direct modulation of TF activity by metabolic state can also contribute to the large hepatic transcriptional changes observed under HFD. Our data support this idea as we found that among the total 374 TFs quantified, 11% (42) were significantly upregulated in HFD and only 1.3% (5) in CD (Fig 4A, Table S4). HFD nucleus–enriched preparations contained not only increased levels of TFs but also the exclusive presence of 10 TFs, many of them playing key roles in metabolic functions (Fig 4B, Table S4).

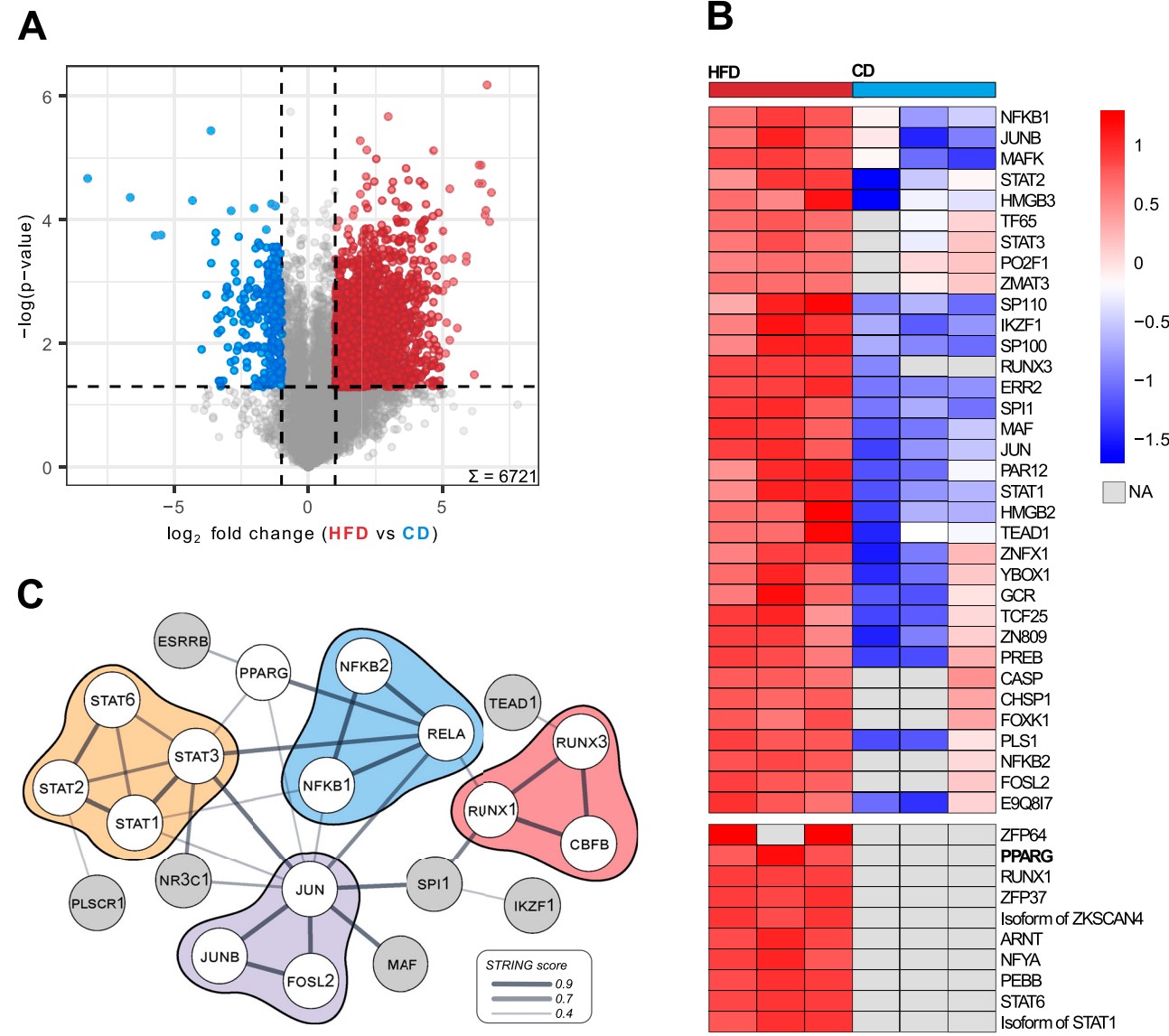

**Figure 4. High-fat diet rewires the nuclear proteome in the mouse liver.**
**(A)** Volcano plot showing the result of a permutation-based $t$ test comparing protein intensities across biological replicates (n = 3) in control (left) and high-fat diet (right). Colored points denote significantly up-regulated proteins with a −log ($P$-value) cutoff of = 1.3 (FDR = 0.05) and a fold change cutoff of $\log_2$ 1 for each diet (CD, blue; HFD, red). **(B)** Heatmap of z-scored protein intensities of quantified transcription factors across biological replicates (n = 3) in CD (right) and HFD (left) with exclusively quantified TFs in HFD in the bottom part. **(C)** String physical subnetwork of up-regulated and exclusive transcription factors in HFD. The TF list was loaded into the STRING app in the Cytoscape platform with *Mus musculus* as a model species and physical network type, and with default search parameters (0.4 confidence cutoff). Singletons were not shown, and only the main network was shown. After network generation, manual arrangement of nodes was performed. STAT proteins were colored yellow; NFkB-related, blue; JUN-related, purple; and RUNX-related, red.

Furthermore, our data reveal that the pleiotropic transcription factor PPARγ is exclusively localized to liver nuclear fractions under HFD conditions (Fig 4B), corroborating previous studies that reported HFD-induced up-regulation of *Pparγ* gene expression from basal levels in homeostatic states (Inoue et al, 2005; Wang et al, 2020; Vesting et al, 2022). These results reveal an additional layer of *Pparγ* regulation in the context of HFD-induced pathology, not only at the transcriptional level (Inoue et al, 2005; Pettinelli & Videla, 2011) but also at the protein and subcellular localization levels. In silico protein–protein interaction analysis using the STRING network database (Szklarczyk et al, 2023) of significantly up-regulated and exclusive TFs in HFD nuclei retrieved a core network of interactions, constituted mainly by the STAT, NFkB, RUNX, and JUN families (Fig 4C). Members of these TF families are known mediators of metabolic homeostasis in the liver, whose function is, often indirectly, reported altered under metabolic challenge (Inoue et al, 2004; Dorn et al, 2014; Dodington et al, 2018; Heida et al, 2021; Bertran et al, 2022). To date, however, very few, if any, reports have investigated the precise modulation of their nuclear accumulation in response to metabolic challenges. Our analysis further offers

precise subcellular quantitative information for some of those factors that have been reported with preferential nuclear localization, as for the transcription factor JUN (Dorn et al, 2014)—a central node in our interaction network. Accurate quantification of nuclear levels is important when assessing treatment options and could be also used for patient stratification and drug response predictions.

Collectively, our results highlight FrozONE's robust capacity to resolve nuclear protein abundance variations in mouse livers under metabolic pathological conditions. FrozONE enables the precise identification and quantification of transcription factors that may serve as pivotal regulators in the molecular mechanisms underlying nonalcoholic steatohepatitis pathogenesis.

## Discussion

The high dynamic range of cellular protein abundance is a well-known challenge for the quantification of low abundant proteins, such as TFs or chromatin modifiers, thus limiting the investigation of nuclear processes using conventional expression proteomics (Wu & Han, 2006). A classical approach to overcome this limitation involves multiple peptide fractionation of individual samples that allows better quantification depth, which holds, in turn, disadvantages such as extensive measurement time and lack of spatial information about protein localization. In contrast, proteomes of enriched organelles (Pisitkun et al, 2004; Rezaul et al, 2005; Takatalo et al, 2006; Wang et al, 2017) display reduced complexity and dynamic range, allowing for a deeper quantification of those low abundant classes of proteins, as well as informing about subcellular protein localization (Christopher et al, 2021). Furthermore, recent advances in mass spectrometry, namely, data-independent acquisition (DIA) (Michalski et al, 2011; Barkovits et al, 2020), have also helped in tackling the dynamic range challenge. Motivated by the potential of subcellular proteomics and the lack of robust and scalable methods that permit large-scale investigation of the nuclear proteome from frozen tissues, we developed FrozONE. Our workflow combines two key aspects: the use of DIA with a quick, reproducible, and scalable nuclear enrichment step from frozen material.

After extensively benchmarking our workflow against conventional density gradient ultracentrifugation–based nucleus enrichment strategies using three mouse tissues, we showed that FrozONE distinguishes itself in terms of quantification depth, reproducibility, and robustness, allowing the comprehensive characterization of nuclear proteomes of distinct organs in a single-shot manner. We indeed obtained a remarkable protein coverage depth, closely similar (80% of total nuclear proteins and 75% of TFs in liver) to the one reported in a nuclear proteome study using SILAC labeling and multiple peptide fractionation (Wang et al, 2017). Our comparison with data obtained from recent subcellular studies (Christoforou et al, 2016; Thul et al, 2017; Go et al, 2021; Cho et al, 2022) showed an overall good overlap (Fig S7, Table S1), despite the diversity of methods and assayed tissues. This overlap was higher, as expected, with the high-purity nuclear proteomes we generated by using FANS from excitatory hippocampal neurons (Fernandez-

Albert et al, 2019). This analysis further pointed to the incompleteness of protein nuclear annotations, which explains the medium percentage of nuclear proteins contained in all tested methods and excludes a high degree of nonspecific non-nuclear proteins in our preparations. We also assessed whether FrozONE conspicuously missed the quantification of nuclear proteins when compared to deep profiling studies that employed extensive sample fractionation strategies, requiring thus much longer preparation and measuring times. By doing this with our liver data, we observed that no obvious groups of proteins with specific nuclear function were not quantified with FrozONE that were present in the other study (Wang et al, 2017). We found instead that these proteins were enriched in annotations from non-nuclear processes/compartments, such as "Lysosome," "Extracellular region," and "Cytoplasm" (Table S3). Given the inherent nature of any nuclear enrichment, it cannot be expected to obtain a grade of purity that resembles the one of an isolation method. Despite FrozONE's efficiency in depleting proteins from non-nuclear organelles (Fig S3C) in similar or better degree than the gold standard method of sucrose gradient, it cannot fully segregate between different possibilities, such as changes in the levels of actual non-nuclear proteins because of abundance change within or relocalization to or from a contaminating organelle. This confounding factor could be further addressed by complementing the quantification of FrozONE-based nuclear-enriched fractions with WCL to better distinguish nuclear-specific from confounder changes.

FrozONE was very efficient in retrieving high TF coverage in a specialized brain area, the hippocampus, compared with analyses employing prefractionation strategies that lead to large numbers of samples and long measurement times (Fig S5E) (Sharma et al, 2015; Liu et al, 2023). Even when compared to a recent characterization of tissue-type restricted TFs (ttrTFs) using immunoprecipitation with consensus sequences (Zhou et al, 2017), we obtained 20–40% more TFs overall and similar tissue-specific abundance of commonly quantified proteins (Fig S4B).

In addition to demonstrating the applicability of our workflow to characterize tissue- and brain area–specific nuclear signatures in healthy conditions, we tested the potential of FrozONE to uncover changes driven by pathological conditions. To this end, we employed our workflow to interrogate nuclear liver signatures in a disease model of diet-induced NASH. FrozONE resolved more nuclear proteins compared with other proteomics studies of HFD livers involving multiple fractionation and ultracentrifugation (3,570, 51% versus 1,891, 38%, respectively) (Liu et al, 2017). Even when compared to a study employing organellar fractionation of HFD livers (Krahmer et al, 2018), FrozONE managed to retrieve 84% of proteins assigned to the nucleus as the main compartment. In our diet intervention study, FrozONE revealed that a high-fat diet is capable of reshaping the abundance of liver nuclear proteins, in particular TFs, some of them already reported to be important for the initiation and development of this pathology (Kern et al, 2018). FrozONE captured overall a higher number of differentially abundant transcription factors in HFD compared with previous proteomics experiments (Zhi et al, 2022). Our data exposed a cluster of transcription factors whose liver nuclear abundance is strongly affected by HFD. These included STAT, JUN, NFkB, and PPARγ families, all reported, often indirectly via motif analysis of RNAseq

or their own gene expression, altered in NASH (Dorn et al, 2014; Grohmann et al, 2018; Kern et al, 2018). In addition, our nuclear proteome profiling also uncovered nuclear changes in metabolic enzymes that link organelle metabolic state to the epigenome and in several therapeutic targets already used in the context of NASH, such as PPARγ, farnesoid X receptor (FXR), thyroid hormone receptor (THR), and pregnane X receptor (PXR) (Trauner & Fuchs, 2022; Umemura et al, 2022). Together, our data indicate that FrozONE performs well with organs whose composition is altered, such as fatty liver, and that it can compete with other organellar fractionation strategies that are resource-intensive and time-consuming.

Collectively, we present FrozONE as a reproducible workflow to comprehensively characterize nuclear proteomes in a short time and without the need for fresh tissue processing and ultracentrifugation, thereby simplifying experimental logistics and increasing scalability. Furthermore, the successful application of FrozONE to both healthy and pathological tissues underscores its potential for diagnostic purposes and the design of therapeutic strategies. The scalability and efficiency of FrozONE facilitate the analysis of large cohorts using minimal amounts of stored frozen samples, offering significant applicability for clinical research. We believe FrozONE can open new avenues for the molecular characterization of nuclear processes in health and disease using biobank available material.

# Materials and Methods

### Animal experiments

Organs were isolated from male C57BL/6N mice of 8–10 wk of age that were maintained in individually ventilated cages under a 12-h light/dark cycle and were provided with sterile food and water under specific-pathogen-free conditions. All procedures were performed in accordance with the European law regarding protection of animal welfare and with approval by the local government authorities (animal facility registered for breeding and use of animals for scientific purpose KVR-I/221-TA166/22_03-06).

The animal diet intervention study was carried out under permission and in compliance with animal protection protocols approved by local government authorities (81-02.04.2022.A396; Bezirksregierung Köln). Male C57BL/6N animals were purchased from Charles River at 4 wk of age. Animals were housed in individually ventilated cages (IVCs) with a 12-h light/dark cycle at 22–24°C. Animals had ad libitum access to water and a normal control diet (13% kcal from fat and 42% kcal from carbohydrates, ssniff E15767-0403; ssniff-Spezialdiäten GmbH). The condition group received a high-fat diet (40% kcal from fat, 42% kcal from carbohydrates, and 2% cholesterol, E15766-3402; ssniff-Spezialdiäten GmbH) from 6 wk of age for 10 wk before euthanasia by cervical dislocation at 16 wk of age.

For FANS experiments, Sun1-GFP x Camk2a-creERT2 mice with a C57BL/6J genetic background were maintained and bred under standard conditions, consistent with Spanish and European regulations. The generation of CaMKIIα-creERT2 ([Erdmann et al, 2007];

from the European Mouse Mutant Archive EMMA strain 02125) and SUN1-tagged mice ([Mo et al, 2015]; stock no. 021039; Jackson Laboratory) has been previously described. All the protocols for animal experimentation were approved by the Animal Welfare Committee at the Instituto de Neurociencias, the CSIC Ethical Committee, and the Dirección General de Agricultura, Ganadería y Pesca de Generalitat Valenciana. In detail, mice were maintained under specific-pathogen-free (SPF) conditions within the Animal House at the Instituto de Neurociencias (CSIC-UMH), in a 12-h light/12-h dark cycle (7:00 AM to 7:00 PM) at 20–24°C and controlled humidity (40–60%) with free access to food and water. At 2 mo old, tamoxifen (20 mg/ml dissolved in corn oil; Sigma-Aldrich) was administered intragastrically, five times on alternate days, to produce the recombination of the floxed alleles and the expression of the Sun1 reporter on the nuclear membrane.

### Tissue collection

Mice for the diet intervention study were euthanized 6 h after lights on. All other animals for centrifugation-based nucleus isolations were euthanized during light hours (between 2 and 6 h after lights on). All animals were euthanized by cervical dislocation. Complete organs were dissected except for the brains, where over a culture plate olfactory bulbs and cerebellum were removed to isolate forebrains. These were further referred to as Brain. To isolate the hypothalamus, brains were placed ventral side up and the hypothalamus was bulged out by gentle pressing with tweezers and then pulled off. To isolate the hippocampi, brains were split into two hemispheres using a scalpel and the hippocampi were scooped out using spatulas as previously described (Bin Imtiaz & Jessberger, 2021) and combined for further processing. After dissection, whole organs were split for fresh preparation, in which they were immediately processed for nucleus enrichment, and for frozen preparation, where they were snap-frozen in liquid nitrogen right after splitting and kept frozen at –80°C until processing. Mice for FANS experiments were euthanized by cervical dislocation 1 mo after recombination. Hippocampi were dissected and immediately frozen with dry ice to preserve the integrity of the tissue.

### Centrifugation-based nucleus enrichment

#### *Sucrose gradient*
Solutions were prepared ahead of time and kept at 4°C until the day of processing. Both buffers had the same composition except for the sucrose molarity (cushion buffer 2.05 M, homogenization buffer 2.2 M). Common elements were as follows: 10 mM Hepes, pH 7.6, 15 mM KCl, 2 mM EDTA, 0.15 mM spermine, and 3.2 mM spermidine. On the day of processing, fresh components were added to sucrose buffers: 1:1,000 1 M DTT and cOmplete EDTA-free Protease Inhibitor Tablets (04693132001; Roche) according to the manufacturer's specifications. In ultracentrifugation tubes (cat. no. 355647; Beckman Coulter), 3.3 ml of cushion buffer was added and kept on ice. To culture plates, 700 µl of homogenization buffer was added over ice, and tissues were preminced thoroughly with scalpels until a fine degree of mincing was achieved. Using self-made wide 1,000-µl pipette tips, minced tissues were transferred to 7-ml glass douncers (cat. no. 357542; Wheaton) labeled previously with a 1.3-ml marking

on ice and filled with HB until reaching the 1.3-ml mark. Another 1.3 ml of 1x TBS buffer was added, and tissues were dounced first with 10 strokes with loose pestle "A" followed by 15 strokes with tight pestle "B." After homogenization, lysates were filtered through 40-$\mu$m cell strainers (CC8111-0042; Starlab) on falcon tubes and centrifuged for 2 min at 100$g$ at 4°C. Filtered homogenates were transferred to falcons containing 8.3 ml homogenization buffer and mixed by inverting. Homogenates were then layered on top of the sucrose cushion very slowly and carefully with "passive" force with a serological pipette, to ensure proper layering and no mixing between sucrose solutions. Samples were centrifuged for 40 min at 100,000$g$ (XL-90, Rotor SW 40 Ti; Beckman Optima) at 4°C. After ultracentrifugation, a layer of fat could be observed on top of the tube and was wiped out with a tissue. Sucrose solutions were aspirated dry until a small white nucleus pellet could be observed on the bottom.

### Iodixanol gradient

Stable 6x homogenization buffer (30 mM CaCl2, 18 mM Mg(Ac)$_2$, and 60 mM Tris–HCl, pH 7.8) was prepared ahead of time and kept at 4°C. On the day of the experiment, first "unstable" 6x homogenization buffer was prepared (stable 6x homogenization buffer and 1 mM $\beta$-mercaptoethanol), and subsequently, 1x "unstable" homogenization buffer (1:6 unstable 6x homogenization buffer, 320 mM sucrose, 0.1 mM EDTA, 0.1% NP-40, and cOmplete EDTA-free Protease Inhibitor Tablets) was prepared. Frozen tissues were brought to ice and left soft/defrost for 10 min. In the meantime, 5.4 ml of 29% iodixanol solution (1:6 unstable 6x homogenization buffer, 160 mM sucrose, and 29% iodixanol) and 4 ml of 1x homogenization buffer were added to UC tubes and 7 ml glass douncer, respectively, and both were left on ice. Soft/defrosted tissues were prepared the same way as above-mentioned under *Sucrose gradient*. After filtering, 3.6 ml of homogenate was taken and mixed thoroughly with equal parts of 50% iodixanol solution (1:6 unstable 6x homogenization buffer, 50% iodixanol) to make it 25%. Then, the 25% iodixanol/homogenate mix was layered very slowly and carefully on top of the 29% iodixanol solution, where a clear separation between the two phases should be achieved. Samples were centrifuged for 30 min at 10,000$g$ (XL-90, Rotor SW 40 Ti; Beckman Optima) at 4°C. Gradient solutions were aspirated dry until a small white pellet could be observed on the bottom (nuclei).

FrozONE workflow using adapted Nuclei EZ Prep protocol was as follows:

1. Dissect frozen tissue pieces (~150 mg for whole tissues, for hippocampus, and for hypothalamus whole tissue) and transfer into 2-ml tubes.

2. Add one stainless steel ball in each tube and 1 ml of cold Nuclei EZ lysis buffer (NUC101; Sigma-Aldrich) supplemented with protease inhibitors as in the previous methods.

3. Homogenize 100 mg frozen tissue using the TissueLyser II (QIAGEN)* at 30 Hz for 30 s in the case of the brain, 20 s for the kidneys, and 15 s in the case of liver in tube holders previously cooled at 4°C. Inspect visually homogenates to ensure proper homogenization. After homogenization, empty tubes over 40-$\mu$m cell strainers (CC8111-0042; Starlab) on top of 50-ml tubes and centrifuge at 100$g$ for 2 min at 4°C.

4. Tap tubes slightly to dissociate any potential pellets and transfer to 1.5-ml tubes sitting on ice.

5. Incubate for 5 min in ice, then centrifuge at 500$g$ for 5 min at 4°C.

6. Remove the supernatant by aspiration carefully with a vacuum pump with a 200-$\mu$l pipette tip at the end.

7. Dissociate aspirated pellets by moving tubes across a tube rack 10 times.

8. Resuspend pellets with 1 ml cold Nuclei EZ lysis buffer and mix by inverting three times.

9. Repeat steps 6–9 three times and remove the supernatant to finish the nuclear enrichment.

*Hippocampus and hypothalamus were homogenized in the glass douncer instead.

### FANS for proteomics

To isolate the nuclei for sorting, frozen hippocampi were mechanically disaggregated in a 2-ml Dounce homogenizer (Sigma-Aldrich) containing 1 ml nuclear extraction buffer (NEB) whose composition is as follows: 250 mM sucrose, 25 mM KCl, 5 mM MgCl2, 20 mM Hepes–KOH, 65 mM $\beta$-glycerophosphate, 0.5% IGEPAL CA-630, 0.2 mM spermine, 0.5 mM spermidine, and protease inhibitors (cOmplete EDTA-free, Roche). Hippocampi were disrupted 15 times with each pestle (A and B). Then, the homogenized solution was filtered with 70-$\mu$m Nylon filters (Corning Falcon) to remove cellular debris. Afterward, nuclei were transferred to 2-ml Protein LoBind Eppendorf tubes and stained with 0.01 mM of DAPI for 10 min at 4°C. To dilute the nuclei and improve the sorting, 1 ml of phosphate buffer solution 1X with additives (0.2 mM spermine, 0.5 mM spermidine, proteinase inhibitors, and 0.01 M MgCl$_2$) was added to the solution up to a final volume of 2 ml. DAPI- and GFP-positive nuclei (neurons) were isolated by FANS in a flow cytometer (FACSAria III; BD Bioscience).

Sorted nuclei were collected in 1.5-ml Protein LoBind Eppendorf tubes filled with 200 ml of nuclear isolation buffer (NIB), whose composition is as follows: 340 mM sucrose, 25 mM KCl, 5 mM MgCl2, 20 mM Hepes–KOH, 65 mM $\beta$-glycerophosphate, 0.2 mM spermine, 0.5 mM spermidine, and proteinase inhibitors. Nucleus morphology and integrity after sorting was assessed with a fluorescence microscope. All buffers used during this protocol (NEB, NIB, and PBS) were prepared using autoclaved material not washed with soaps to avoid contamination with detergents and then filtered with 0.22-$\mu$m sterile filters (VWR). Nuclei were centrifuged for 10 min at 1,500$g$ (4°C), and the supernatant was removed. Immediately after, nuclei were snap-frozen in liquid nitrogen.

### Proteome sample preparation

Dry nucleus pellets were dissociated by moving tubes across EPPI racks. Afterward, proteomes were prepared based on the EasyPhos method for proteomes (Humphrey et al, 2018). Buffers were dissolved in Milli-Q water unless stated differently. In detail, 300 $\mu$l of lysis buffer (2% SDC [sodium deoxycholate, 30970; Sigma-Aldrich], 100 mM Tris, pH 8.5) was added to pellets and samples were immediately boiled at 95°C for 5 min with agitation (1,500 rpm, Eppendorf ThermoMixer C) to inactivate phosphatases and

proteases. After boiling, samples were cooled off in ice for 2 min before sonication for 45 cycles 30-s on and 30-s off at maximum output (Bioruptor Plus). After sonication, protein concentration was determined by BCA assay (Pierce BCA Protein Assay Kits). Aliquots of 10 $\mu$g protein were taken, and volumes were equilibrated with leftover lysis buffer. Afterward, 10x alkylation/reduction buffer (100 mM TCEP-HCl, 400 mM 2-chloroacetamide at pH ~7 adjusted with KOH) was added 1:10 and incubated for 5 min at 45°C with agitation (1,500 rpm, Eppendorf ThermoMixer C). Samples were subsequently digested simultaneously with trypsin (cat. no. T6567; Sigma-Aldrich) and LysC (cat. no. 129-02541; Wako Chemicals) at a 1: 100 protein:enzyme ratio overnight at 37°C with agitation. Tryptic digests were acidified by adding equal amounts of SDB-RPS loading buffer (2% TFA, 98% isopropanol). Acidified digests were then loaded in two steps of 150 $\mu$l on in-house-made three-layer SDB-RPS StageTips (CDS Empore SDB-RPS Extraction Disks [Rappsilber et al, 2007]) supported in 3D-printed adaptors for 2-ml tubes and centrifuged for 8 min at 1,500$g$ or until liquid is completely through. After loading, StageTips were washed with 100 $\mu$l of SDB-RPS loading buffer and centrifuged for 5 min at 1,500$g$. After this, samples were washed with 100 $\mu$l of SDB-RPS wash buffer (0.2% TFA, 5% acetonitrile (ACN)) and centrifuged for 5 min at 1,500$g$. For elution, right before usage, $NH_4OH$ was added to the SDB-RPS elution buffer (80% ACN, 0.3125% $NH_4OH$). StageTips were placed on a 3D-printed 96-well centrifuge adapter on PCR tubes. After this, 60 $\mu$l of SDB-RPS elution buffer was added and samples were centrifuged for 5 min at 1,500$g$ or until completely through. Immediately afterward, tubes were placed on an evaporative concentrator (Eppendorf Vacuum Concentrator Plus) and centrifuged till dryness for 30 min at 45°C. Then, samples were resuspended with 10 $\mu$l A*-buffer (20% ACN, 0.1% TFA) and kept frozen at –20°C until measurement.

## LC-MS/MS

Two hundred nanograms of desalted peptides was loaded into in-house–packed 50-cm reversed-phase columns (75 cm diameter, 1.9 mm C18 ReproSil particles, Dr. Maisch GmbH) with the EASY-nLC 1,200 system (Thermo Fisher Scientific) at 60°C. Livers and brains for method comparison were measured with a gradient length of 120 min (5–30% buffer B for 105 min, to 95% for 5 min, constant for 10 min) with a flow rate of 300 nl/min. Kidneys and the livers for the pathological study were measured with a gradient length of 90 min (5–30% buffer B for 75 min, to 95% for 5 min, constant for 10 min) with a flow rate of 300 nl/min (Vanquish HPLC, Thermo Fisher Scientific), which were found to produce highly comparable data to the longer one. Elution and separation of peptides were achieved using a binary buffer system between buffer A (0.1% formic acid, MS-grade water) and buffer B (80% ACN, 0.1% formic acid, MS-grade water). Eluted peptides were electrosprayed into an Exploris 480 mass spectrometer (MS) (Thermo Fisher Scientific). Full scans (MS1) were set with Orbitrap resolution 120,000, scan range 350–1,000 m/z, maximum injection time 45 ms, data type "Profile," and normalized AGC target to 300%. DIA scans (MS2) were set with Orbitrap resolution 15,000, scan range 350–1,000 m/z, window overlap 1 m/z, normalized AGC target to 1,000%, and maximum injection time to automatic. For DIA isolation window calculation, the median retention time was calculated from previous DDA runs of the

same length and divided by the number of desired data points per peak (n = 6) to obtain cycle time. The number of DIA windows was calculated by subtracting cycle time with MS1 transient time at a resolution of 120,000 (259 ms), and then dividing by MS2 transient time at a resolution of 15,000 (32 ms). The isolation window was then calculated by rounding the result of subtracting scan range values (1,000-350 m/z) and dividing it by the number of DIA windows.

## Data analysis

MS raw files were processed with DIA-NN (Demichev et al, 2020) (version 1.8.2 beta 22) with all tissues together using default parameters unless stated: spectral library generation ON with mouse FASTA files UP000000589_10090 and UP000000589_1009 0_additional (obtained on 27/10/2022). The peptide length range was set for 7–35 and precursor m/z range 350–1,000. Protein inference was set to "Protein names (from FASTA)" and quantification strategy to "Composite (high precision)." After quantification, protein group output files "pg_matrix.tsv" were imported into the Perseus platform (version 2.0.10.0). Raw intensities were $log_2$-transformed to make them normally distributed, and proteins were annotated using GO terms (GOBP, GOCC, and GOMF), KEGG database (Kanehisa & Goto, 2000), UniProt keywords, the COMPARTMENTS database (Binder et al, 2014), and AnimalTFDB 4.0 (Shen et al, 2023) for Transcription Factors and Cofactors. We considered proteins as "nuclear" if they were annotated with "Nucleus" in UniProt or COMPARTMENTS keywords or both. To annotate important functional groups of nuclear proteins, we used the following filters: nuclear shuttling ("Nucleus" and "Cytoplasm" in UniProt keywords), chromatin modifiers ("chromatin modification|chromatin remodeling|chromatin assembly| chromatin silencing|chromatin maintenance|chromatin organization" in GOBP or "chromatin remodeling complex|chromatin silencing complex|chromatin assembly complex|chromatin accessibility complex" in GOCC), DNA repair machinery ("DNA repair|base-excision repair|nucleotide-excision repair|double-strand break repair|interstrand cross-link repair" in GOBP), and nuclear transport machinery ("nuclear import|nuclear export|nuclear transport|import into nucleus|protein targeting to the nucleus|protein export from nucleus" in GOBP).

Proteins were considered confidently quantified if they had at least two values among triplicates in at least one group (tissue, method, and/or condition). This filter was applied before each comparison with a different set of groups. The only exceptions to this filter were for PCA, proteins were filtered for 100% valid value quantifications, and for valid value distributions (Fig S1A), no filter was applied.

For reproducibility analysis, Pearson's correlation and CVs were calculated in Perseus and plotted with ggplot. For $t$ test analysis, missing values were imputed by replacing them from a normal distribution (width 0.3, down shift 1.8, separately for each column); they were calculated in Perseus (two-sided, 250 randomizations, FDR < 0.05, S0 = 0.1, and no group preserving during randomizations) and visualized with ggplot.

Exclusive identifications were obtained after applying the valid value filter above-mentioned and averaging $log_2$ intensities

between replicates, and UpsetR was used for visualization. Tissue-up-regulated proteins were determined by imputing missing values from the sample normal distribution (width 0.3, down shift 1.8), followed by multiple-sample testing (ANOVA as indicated in figure legends) in Perseus. Exclusive and up-regulated proteins were combined, and subsequent enrichment analysis was performed using Fisher's exact test in Perseus (BH FDR and $P < 0.02$) taking all quantified proteins as background. Liver and Brain KEGG terms were further filtered to have an intersection size of at least 10 before plotting. Cell-type marker proteins for brain, liver, and kidney were obtained from published literature (Azimifar et al, 2014; Sharma et al, 2015; Sigdel et al, 2020). ggplot2 was used to plot their $\log_2$ protein intensities and overall distributions, and patchwork was used to combine plots. The neurodegenerative disease Venn diagram was created using ggVennDiagram.

For the diet intervention analysis, protein group output files "report.pg_matrix.tsv" were imported into the Perseus platform and data were treated the same way as stated above. For the volcano plot, a valid value filter of two out of three values in at least one group was applied, and then, missing values were imputed by replacing them from a normal distribution (Perseus's default parameters). A t test was performed using Perseus's volcano plot function with default parameters, and the significance threshold was set for an absolute $\log_2$ fold change of 1 and $-\log_{10}$ ($P$-value) of 1.3 (permutation-based q < 0.05). Transcription factors significant in HFD under this threshold were combined with the ones exclusively found in HFD (at least two out of three exclusively in one condition and not the other). Then, missing values were imputed by replacing them with the smallest intensity value found in this merged dataset, to then be row-wise–z-scored and split into the two clusters. After separation, missing values were replaced back to NaN in each dataframe separately, and then, they were merged back together to represent them in the heatmap. For network analysis, gene list of exclusively quantified and up-regulated TFs (defined by ATFDB 4.0) in HFD was loaded into the STRING app (version 2.0.1) in the Cytoscape platform (version 3.10.1), with *Mus musculus* as a model species and physical network type, and with default search parameters (0.4 confidence cutoff). Singletons were not shown, and only the main network was shown. After network generation, manual arrangement of nodes was performed, and the network was exported as a PDF for further visual processing in the InkScape platform (version 1.1.2).

For reporting overlapping quantification with nuclear proteins from indicated literature (Fig S7), Go et al proteins were filtered for those they predict with NMF and SAFE as nucleolus, nuclear body, nuclear outer membrane–ER membrane network, nucleoplasm, chromatin, splicing speckles, nuclear pore, and paraspeckles, Thul et al and Christoforou et al proteins were filtered for nucleus, Cho et al (2022) proteins were filtered for grade 3 (very prominent localization) nuclear proteins, and all nonhuman proteins were matched to mouse orthologs using the Ensembl database before matching by mouse protein group UniProt id and plotting overlaps using UpsetR.

## Data Availability

The mass spectrometry proteomics data have been deposited to the ProteomeXchange Consortium via the PRIDE (Perez-Riverol et al, 2022) partner repository with the dataset identifier PXD050658.

## Supplementary Information

## Acknowledgements

We thank all the members of the Robles' laboratory for critically reading the article. Research work in the group of MS Robles is supported by LMU Munich's Institutional Strategy LMU excellent within the framework of the German Excellence Initiative, the German Research Foundation DFG (INST 86/1800-1 FUGG, RO 5675/1-1, and Project ID 213249687—SFB 1064), and MS Robles and Á Barco are both supported by the RGP0039/2017 from the Chan Zuckerberg Initiative (Grant #: 2024-338475 (5022) GB-1604314). I Bustos-Martínez is the recipient of the contract PRE2021-099569 funded by MICIU/AEI/10.13039/501100011033. Á Barco research is supported by grants PID2020-118169RB-100 abd PID2023-148442NB-I00 funded by MICIU/AEI/10.13039/501100011033 and the European Union.

### Author Contributions

LA Huschet: resources, data curation, formal analysis, validation, visualization, methodology, and writing—original draft, review, and editing.
FP Kliem: resources, data curation, formal analysis, validation, visualization, methodology, and writing—original draft, review, and editing.
P Wienand: resources.
CM Wunderlich: resources.
A Ribeiro: resources.
I Bustos-Martínez: resources.
Á Barco: resources.
FT Wunderlich: resources.
M Lech: resources.
MS Robles: conceptualization, resources, supervision, funding acquisition, methodology, project administration, and writing—original draft, review, and editing.

### Conflict of Interest Statement

The authors declare that they have no conflict of interest.

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
