## [Reviewer comments · Life Science Alliance]

Life Science Alliance

FrozONE: Quick Cell Nucleus Enrichment for Comprehensive Proteomic Analysis of Frozen Tissues

Maria Robles, Lukas Huschet, Fabian Kliem, peter Wienand, Claudia Wunderlich, Andrea Ribeiro, Isabel Bustos-Martinez, Angel Barco, Thomas Wunderlich, and Maciej Lech

DOI: <https://doi.org/10.26508/lsa.202403130>

Corresponding author(s): Maria Robles, Ludwig-Maximilians-Universität München

Review Timeline:	Submission Date:	2024-11-08
	Editorial Decision:	2024-11-11
	Revision Received:	2024-11-27
	Editorial Decision:	2024-12-02
	Revision Received:	2024-12-02
	Accepted:	2024-12-03

Transaction Report:

Response to reviewer comments

(Huschet, LA; Kliem FP et al.)

Reviewer #1:

The manuscript by Huschet et al. introduces a new experimental protocol for performing nuclear-cytoplasmic fractionation. Specifically, this method allows for nuclear isolation from frozen tissues with equivalent data quality, and a faster experimental workflow, compared to current gold standard methods. The effectiveness of this method was demonstrated on a set of diverse tissues, brain, liver, and kidney, suggesting that the method may be suitable for a wide range of tissues, both frozen and fresh. The authors demonstrate the reproducibility of the method, showing both CVs lower or comparable to gold standard methods and close localization of replicates within a PCA plot. Finally, the authors use their method to explore a biological question, diet-induced NASH, showing rewiring of nuclear proteome compositions in response to diet.

We thank the reviewer for their positive evaluation of our manuscript and the recognition of our method's effectiveness and reproducibility across diverse tissues. We have now worked on the comments raised by this reviewer and tried to add clarity to all parts in the text and figures.

Comment 1: At the bottom of page 5 the authors discuss how they used nuclear cell type markers to demonstrate that their method was capturing different cell types within a given tissue, showing that up to 89% of the markers they selected were observed. This result needs more detail. For example, cell types are often represented by multiple markers, thus the authors should state how many cell types they are looking at using their selected markers. It makes a difference if the authors were looking for only three cell types and detecting two of these (clearly not the case here) versus looking for 100 cell types and finding 60 of these. This information is needed so that readers can get a better understanding of how well the method is representing different cell types within the tissue.

We thank the reviewer for pointing this out, indeed we missed to state the number of cell types defined by using previous published datasets. Taken all the studies together we classified a total of 11 cell types defined by a specific number of markers. Our data contained a high percentage of specific markers from all the 11 cell types. We have now incorporated this information in the revised version of the manuscript (highlighted on page 5, "To this end, ...").

Secondly, if possible, an analysis of the detected cell type markers should be done to determine if the detected cell types are found in proportions that are roughly expected. Obviously, getting an exact calculation might be challenging, especially for small differences. However, are there any cell types that one would perhaps expect to see at say a 3:1 ratio in the tissue whose cell type markers have a somewhat similar ratio in the data? Such an analysis would be helpful in understanding if the method has a bias towards a particular cell type.

This is a very good point and we agree that it would be very interesting to be able to predict cell type proportions by using our quantitative data from the different cell type markers. However, cell type marker differences may not precisely reflect cell type ratio within the individual tissues as it could be affected by the overall protein level of those markers within the nuclear proteome of the individual cell type. For example, markers of cell type 1 (of low proportion) with high abundance may display similar protein intensities to markers of cell type 2 (high proportion) with lower abundances. Nevertheless, and even considering this pitfall, we looked in detail into our data and associate protein intensity fold changes of markers to the cell type proportion reported in the three tissues and found an overall congruent trend.

In the kidney, glomerular cells are reported to be 5% of the kidney weight¹ and proximal tubular cells around half of the protein mass². This proportion is recapitulated by the fold change of non logarithmic intensities (3.2; see **Fig 2D** panel) of protein markers of proximal tubular cells vs glomerular cells.

Excerpt of Fig 2D.

In the brain, the reported proportions are: ~10% microglia³, 10-20% astrocytes⁴, ~20% oligodendrocytes and ~50% neurons⁵. This distribution correlates overall with the fold change intensity values obtained when

comparing markers of astrocytes and neurons to the least abundant cell type microglia (3.5 and 6.1 respectively). The fold change with oligodendrocytes is however higher than expected (16.1), most likely due to the high standard deviation caused by one of the markers (as seen in **Fig 2D**, also below). We observed the same trend when performing an alternative comparison using the protein levels of an exclusive cell marker, NeuN and OLIG2, for neurons and oligodendrocytes, respectively⁵ (see **Fig R1C1** here and now **Fig S3A** in our revised version).

Excerpt of Fig 2D.

Figure R1C1. Normalised, log₂ transformed intensities of the neuron marker protein RFOX3 (NeuN) and oligodendrocyte marker protein OLIG2, averaged across 3 brain FrozONE replicates (now **Fig S3A**).

Similarly, our quantitative data of nuclear annotated liver cell type markers also correlate with reported cell type proportions. Thus, we obtained fold change intensity values of 1.4, 3.2, 3.7, 13.9 when comparing markers of cholangiocytes (CHC 3-5%⁶), Kupffer cells (KF 15%⁷), hepatocytes (HC 80%⁸) and liver sinusoidal endothelial cells (LSEC 15-20%⁹) to hepatic stellate cells (HSC 5-10%¹⁰).

Excerpt of Fig 2D.

We have now incorporated these comparative analyses in the revised manuscript under a highlighted paragraph in page 5 ("In addition to its great depth ...").

2: Currently, the writing of the NASH experimental section does not sufficiently highlight the new biological insight. This section seems focused on highlighting the fact that the authors are reconfirming many aspects of NASH biology using their method. The TF analysis focuses on the JUN centered network, which the authors note is known to have upregulated gene expression. The authors also mention PPAR γ , which has also been

previously identified as upregulated in HFD. It would be good if the authors could also emphasize some of the new findings, and/or do some additional analysis on these data.

We thank the reviewer for this comment. As our manuscript was submitted as a method article, we intended to use the NASH experimental section to show the applicability of FrozONE to a pathological condition. Our overall aim was therefore to demonstrate that our analysis can capture in a collective and comprehensive manner known molecular changes caused by high fat diet that have been reported by a set of individual studies using different technologies. Additionally, and perhaps not sufficiently indicated in our first version of our manuscript, our data provided new insights to previously reported data. We precisely quantified, under both diet conditions, the nuclear levels of transcription factors whose HFD-driven impaired activity is often indirectly predicted by i) their own mRNA levels or ii) gene enrichment analysis^{11,12}.

Nevertheless, we have incorporated the suggestions from the reviewer and have deepened our analysis of our NASH data by performing and including an enrichment analysis of those significantly upregulated nuclear proteins in each diet. In accordance with the altered metabolic state because of a NASH diet, several Uniprot metabolism-related annotations were significantly enriched, including Lipid- and Fatty acid biosynthesis, Lipid metabolism, NAD and NADP (Table S4). We specifically highlight the NASH induced nuclear levels of several enzymes involved in Acetyl-CoA metabolism. As Acetyl-CoA is utilized as an acetyl group donor for histone acetylation, our data indicate that higher nuclear accumulation of those enzymes could be a molecular underpinning driving the reported HFD-driven changes in histone acetylation levels (Arias-Alvarado et al, 2021). In particular we observed HFD-associated higher levels of ATP Citrate Lyase (ACLY), a key enzyme mediating the conversion of citrate to Acetyl-CoA in the nucleus, pointing to an increase production of nuclear Acetyl-CoA levels that could be subsequently stored in form of histone acetylation (Boon et al, 2020). We can thus speculate that metabolic state directly impacts ACLY nuclear accumulation, connecting, in turn, metabolism with the epigenome. Moreover, within these enriched pathways a transcription factor and cofactor can be found, indicating that not only changes in proteins directly involved in such processes are taking place, but also changes around transcription (highlighted on page 8, "Furthermore, we observed that nuclear annotated proteins...").

We consider that the accurate determination of nuclear protein levels of those transcription factors, in particular nuclear receptors often used as therapeutic targets in HFD¹³ is of translational applicability. For example, assaying nuclear levels of those drug targets in patient biopsies would be of potential use for cohort patient stratification in therapeutic approaches and/or to predict treatment outcomes. This is now included in page 9 of the revised manuscript "Members of..."

Minor Comments

1: For supplementary figure 1A, please add labels for at least one of the 6-squared heatmaps. At first, I assumed that the top left corner of the heat maps represented Replicate 1 vs Replicate 1, and therefore should have a correlation of 1. However, I can see that the Liver Sucrose Frozen heatmap clearly does not have a correlation of 1 in that top left box. Thus, it is unclear which replicates are being compared in each square of the heatmaps.

We have now incorporated the recommendation of this reviewer and modified accordingly the figure labeling replicate numbers.

Reviewer #2:

It is important to be able to dynamically measure the protein constituent parts of the nucleus, as major cellular processes are enacted within its environs. Moreover, aberrant subcellular location underlies many pathological states such as ALS, FTD and Huntington's and Alzheimer's diseases. To date, the characterisation of the nuclear proteome has been the focus of a number of different methodologies. Lower throughput methods focus on imaging for example using a fluorescently labelled antibody raised against nuclear proteins in interest. Other approaches do not require prior knowledge of nuclear proteins and are used often to discover novel components of the nuclear proteome or to interrogate the whole nuclear proteome, rather than protein by protein. These methods broadly fall in three categories:

1. Nuclear enrichment, using biochemical fractionation or affinity purification
2. Proximity labelling methods, where an enzyme such as ascorbate peroxidase or biotin ligase (or their derivatives) are trafficked to the nucleus either through fusion to a nuclear protein or via a nuclear import sequence, and in situ, form activated biotin that reacts with proximal proteins that can be enriched and identified by mass spectrometry. These methods are restricted to samples that can be easily genetically manipulated.

3. Protein correlation profiling methods, where cell wide localisation of the proteome is possible through biochemical fractionation and classifying the distribution of proteins amongst fractions based on the distributions of marker proteins, in this case, nuclear markers.

All the above approaches have been used to map the nuclear proteome to date either as discreet nuclear datasets, or data within the context of the subcellular distribution of the proteome within the entire cell. The method described by the authors, aims to improve of the first category above by providing a robust, reproducible and scalable method, FrozONE to create a nuclear enriched sample. Moreover, the method is able to deal with frozen samples, which to date have not been an optimal starting material for density centrifugation methods. Enriched nuclear fractions have typically been achieved using gradient density centrifugation methods, which are limited in their scalability accessibility, due to the requirements for ultracentrifugation.

Another issue with nuclear fractionation methods, has been leakage of soluble nucleoplasm proteins during fractionation, resulting in incomplete data. The authors benchmark the FrozONE method against other more traditional gradient fractionation approaches and also determine the differences in the nuclear proteome of three mouse tissues, brain, liver and kidney. Finally, to demonstrate the applicability of the FrozONE method to studies characterizing nuclear proteins in differential settings, they describe changes in the liver nuclear proteome associated with nonalcoholic steatohepatitis induce by a high-fat diet induce. They reveal a plethora of related changes in protein abundance and highlighting the potential use of the method in conjunction with clinical samples. The FrozONE method holds great potential, as it is scalable, as it does not require access to low throughput centrifugation and, from the data shown, is reproducible.

We appreciate the reviewer's comprehensive summary and insightful remarks on the significance of measuring the nuclear proteome and the challenges associated with existing methodologies. We are grateful for the recognition of our FrozONE method's potential to improve nuclear fractionation, particularly its scalability, reproducibility, and effectiveness with frozen samples, as demonstrated with our benchmarking experiments.

I have several significant concerns that, in my opinion, preclude the publication of this method as presented. The first is that the method employs a commercial kit, which to date, has been used for nuclear preparations upstream of RNAseq workflows. FrozONE is very reliant on the kit and it is not clear how it works, if there are any alternatives or what would happen if the manufacturers stopped selling it.

We understand the reviewer's concern about the requirement of a commercial kit for the FrozONE method, however, this method is not an exception, at least in the proteomics field, employing commercial material. There are several examples of widely used methods and protocols in the community that rely on commercial products, from protein quantification kits (such as Tandem mass tags (TMT) and SILAC metabolic labeling), material to enrich post-translational modifications (antibody-based- acetylation, ubiquitination, tyrosine phosphorylation- and TiO₂ bead-based global phosphopeptide enrichment) to general instrumentation

(analytical HPLC columns, etc). Overall, the requirement of commercial products neither diminish the value nor the general use in the community of those protocols. Moreover, the manufacturer is well known and has a strong track record of consistent availability, technical support and quality of the products. We therefore do not consider this a crucial bottleneck in the use of our protocol in a short or long term perspective.

Secondly, the authors judge the success of the method based on how many proteins it returns. It is clear from figure 1B that half the proteins are not nuclear. I would like to have seen a better representation than of the overlap of proteins identified for each method, and also assessment of the 'contaminating' proteins, in other words, those proteins identified by FrozONE which are not annotated as nuclear in UniProt or COMPARTMENTS.

We agree with the reviewer that it would be informative to report the data in a different manner than just by merely reporting yielded protein numbers. We have now done this by precisely looking at the overlap of individual proteins quantified with FrozONE and sucrose fresh. We observed that 73-87% of the total quantified proteins are common to both methods and only 13-27% are exclusive to one of the methods. Overall, the fraction of nuclear annotated proteins (UniProt + COMPARTMENTS) are similar within the method-exclusive proteins, indicating that both methods equally enriched nuclear proteins (based on annotations, see further analyses on this topic below on page 9). This data is now included here as **Fig R2C1** and in **Fig S2A** of the revised version of our manuscript.

Figure R2C1. Upset plot showing the overlap in protein identities between FrozONE and Sucrose fresh quantifications in Brain, Liver and Kidney, subdivided into all proteins in light colored bars and nuclear annotated proteins in saturated bars. Total numbers of proteins obtained with each method in each tissue is indicated on the colored bar plot on the left side of the upset plots. Note that the total number of proteins is slightly higher than in **Fig 1B** as it shows proteins quantified in $\frac{2}{3}$ replicates rather than in all 3 replicates (**Fig 1B**).

Moreover, as suggested by the reviewer, we also assessed in detail the quantification robustness and reproducibility of proteins not annotated as nuclear from FrozONE preparation, by using identification reproducibility and coefficient of variation of protein intensities, respectively. First, we observed that most proteins (total, nuclear and non-nuclear annotated) are quantified in the three replicates of all methods for all tissues. If non-nuclear annotated proteins will be randomly quantified due to sporadic contamination during the enrichment we would not expect to have them consistently presented in all replicates, as we do observe and display in **Fig R2C2** below (Revised Fig S1A). When comparing FrozONE and sucrose fresh methods we see a high correlation of quantification robustness.

Figure R2C2. Barplot showing the number of proteins (total, nuclear and non-nuclear annotated) quantified in 1, 2 or 3 replicates from each method applied to each tissue.

Secondly, by comparing the coefficient of variation (CVs) of protein intensities for non-nuclear annotated proteins across methods we observe that FrozONE consistently yields the lowest values like what we have shown for all proteins and nuclear annotated proteins (Fig R2C3 right panel, now included as Fig S1C). This indicates a lack of randomness in the quantification of non-nuclear proteins, which are overall consistently detected (as indicated above) with minimal variation in intensities. Thus, even if considering non-nuclear proteins as contaminants of the enrichment, rather than incompletely annotated proteins (as it seems the case based on our new analysis, see below in page 9), they will not affect the robustness of the results when comparing nuclear proteomes of different experimental conditions using FrozONE. Therefore, we

conclude that, as stated in the manuscript, FrozONE is a strong competitor to conventional and more tedious nuclear enrichment methods in regard to quantification robustness and reproducibility without an increased rate of variability of non-nuclear annotated proteins.

Figure R2C3. Box plots of coefficients of variation of log₂-transformed protein intensities between 3 replicates for each tissue prepared with the indicated method, for all proteins, nuclear annotated and non-nuclear annotated proteins.

These additional analyses are now included in the revised manuscript and highlighted on pages 3-4.

To allay my concerns about nuclear leakage, I would also have welcomed analysis on representation of subnuclear compartments for example, chromatin associated, nucleoplasm, nuclear membrane and nucleolar, as well as an analysis of which nuclear proteins are conspicuous by their absence from the data.

We acknowledge the point made by the reviewer and carefully review our data in different ways to clarify their concerns. We now repeated our analysis of CVs and missing value distributions including the nuclear terms mentioned by the reviewer. Overall, we observed very similar values and proportions as the total nuclear annotated proteins, indicating a lack of specific loss of proteins from a particular nuclear annotation, including nucleoplasm (see Fig R2C4). Therefore, while we cannot completely rule out a minimal degree of nuclear leakage, from our data we can conclude that if this would be the case i) the leakage will be of minimal degree with FrozONE compared to other nuclear preparation methods and ii) the quantification robustness and reproducibility of different nuclear protein subgroups in FrozONE samples are very stable, equally or more than in samples from other conventional methods.

Figure R2C4. Top left, as **Fig R2C2**, barplot showing the number of proteins quantified in 1, 2 or 3 replicates of every combination of tissue and method. The rows subdivide the proteins into the GOCC categories: nuclear membrane, nucleoplasm, nucleolus or chromatin binding. Top right, as **Fig R2C3**, box plots of coefficients of variation of log₂-transformed protein intensities between 3 replicates for each main tissue and method. The plots are subdivided to show the CV distributions of all proteins, nuclear annotated, non-nuclear annotated proteins and proteins annotated with the GOCC categories nuclear membrane, nucleoplasm, nucleolus or chromatin binding. Bottom left, Barplot representing enriched (Fisher exact test, FDR = 0.05) GO cellular compartment (GOCC) terms from proteins not quantified by FroZONE that are present in Wang et al's¹⁴ nuclear proteome.

We further analyzed our liver nuclei data and compared it to a study employing density gradient with fresh tissue and peptide fractionation that reported the highest depth and coverage so far of a mouse liver nuclear proteome¹⁴. By specifically looking at those proteins present in this study and not in our FroZONE liver samples we do not detect an obvious overrepresentation of any group of proteins with specific nuclear function. This will be the case if our method produces nuclear leakiness and/or is not sensitive enough to quantify low abundant nuclear proteins. Instead, we found that those proteins missing in our FroZONE preparations are enriched in non-nuclear processes/compartments, such as "Lysosome", "Extracellular region" and "cytoplasm" (**Fig R2C4**, bottom left), indicating the enrichment efficiency of our FroZONE method. We have included this comparison in the revised version of the manuscript (page 10, "We also assessed whether...").

Finally, I would like to have seen overlay of the data with other cell wide subcellular proteomics data sets such as (Go, Knight et al. 2021) (Thul, Åkesson et al. 2017) (Cho, Cheveralls et al. 2022) (Christoforou, Mulvey et al. 2016). Without these data, the potential users of the method would have no idea what portion of the nuclear is being captured by this method however reproducible it may be.

We thank the reviewer for this suggestion. This point is, in part, answered by the comparison we showed above with the study of Wang et al¹⁴, that employed peptide fractionation to boost identification at the cost of increasing sample number and measurement time. Nevertheless, and to satisfy this reviewer, we also compared our data to the suggested datasets. However, due to the technical pitfalls of their employed methods and the different experimental approaches we consider these comparisons considerably less informative.

Go et al.¹⁵ used a proximity-dependent biotinylation approach in HEK293 cells using 192 subcellular markers to localize 4145 proteins. Of the assigned 1508 proteins to nuclear related compartments with mouse orthologs (1445) we quantified 1136 (79%) in our FroZONE data (only 1088 with Suc fresh; 95% overlap), overall, a

very high coverage considering the different cell types and species. The fraction of nuclear proteins that are not contained in FrozONE datasets are not enriched for nuclear processes but rather lipid metabolism or associated to the endoplasmic reticulum (Fisher's exact test, p-value & BH FDR < 0.02; **Fig R2C5**).

Figure R2C5. Enrichment analysis (Fisher exact test, p-value<0.02, FDR<0.02) of proteins which were identified as nuclear in Go et al 2021 but were not quantified in FrozONE.

Thul et al.¹⁶ used an immunofluorescence microscopy approach to classify 12003 proteins in 22 different human cell lines (www.proteinatlas.org), 6245 of which they assigned to the nucleus and related substructures. This classification overlaps with a high-resolution spatial proteomics map in 76% unique matches (single organelle assignment) and 82% for partial matches, a fraction very similar to the overlap we indicated above between FrozONE nuclei liver proteome and the study of Wang et al.¹⁴. Out of the 4987 FrozONE proteins with human orthologs in the protein atlas dataset, 2869 (58%) were annotated as nuclear. Out of the 5642 human proteins with nucleus as main localization that matched to mouse orthologs, 2374 (42%) were quantified by FrozONE (2458 by Suc fresh; 92% overlap). Again, we consider this a very good overlap considering that protein atlas is based on human cell lines from a variety of tissues.

Cho et al.¹⁷ genetically tagged and detected 1310 proteins with split fluorescence labels that failed to localize proteins residing in organellar lumen. Out of the 503 prominent nuclear human proteins (opencell.czbiohub.org) with mouse orthologs we quantified 347 (69%) with FrozONE (363 with Suc fresh, both 97% overlap).

Christoforou et al.¹⁸ coupled extensive cellular fractionation of mouse pluripotent stem cells with mass spectrometry (hyperLOPIT), which allowed unambiguous assignment of 2855 proteins to discrete organelles and sub-compartments. Of their 693 proteins classified as nuclear, we quantified 611 (88%) with FrozONE (601 with Suc fresh, 99.5% overlap).

We have included all these comparisons in the revised version of our manuscript (page 10) and in Table S1 and **Fig S6A (R2C6)** below for this reviewer).

Figure R2C6. Upset plot showing the overlap of proteins quantified by FrozONE and proteins identified as nuclear in selected cell wide subcellular proteomics data sets (see Table S1). For non-human studies only proteins that could be matched to mouse homologue proteins are shown.

In summary, we observe that each methodology yielded a fraction of exclusive proteins, likely due to the employed method, model organism and tissue. However, and despite the different methodologies and experimental approaches, the overlap of our data, often higher than that from the gold standard sucrose fresh, with those studies is quite remarkable.

Additionally, and to better address the point raised by the reviewer without the bias of different methodologies, we decided to team up with the group of Angel Barco (Neuroscience Institute, Spain) to produce what we consider the current cleanest nuclear preparation using fluorescence activated nucleus sorting (FANS). For this, we employed a mouse model in which a GFP-fusion of the nuclear membrane protein SUN1 is conditionally expressed in CAMKIIa positive neurons in the hippocampus^{19,20}. Sorted SUN1-GFP positive nuclei from mouse hippocampus were used to prepare and measure proteomes using the same acquisition method employed by FrozONE. Our measurement yielded 3864 quantified proteins which despite the high nuclei purity preparation only 57% of them were annotated as nuclear by the comprehensive annotation we employ, indicating the incompleteness of these annotations. When compared to the FrozONE hippocampus proteome we observed that almost 80% of the FANS proteome was covered by FrozONE. As FANS samples only contained neuronal nuclei, we expected that the exclusive FrozONE proteins would be coming from non-neuronal hippocampi cell populations. This is indeed what we observed when we did an enrichment analysis of this set of proteins using brain cell markers based on Sharma et al.²¹, with a concomitant de-enrichment of neuronal markers (**Fig R2C7** here and revised Fig S4B). With this comparison we can therefore conclude that FrozONE proteomes are of good depth and specificity and that in general, nuclear annotations are not the best approach to assess nuclear enrichment and purity.

Figure R2C7. Overlap of proteins quantified with FrozONE in hippocampus and with fluorescence activated nucleus sorting (FANS) using Sun1-GFP in hippocampus and enrichment of cell type markers (Sharma et al 2015) among exclusive FrozONE quantified proteins.

We have included these additional analyses in the revised manuscript on page 6-7 & 10, Fig S4B & S6 and Table S1.

Moreover, the contaminants could lead to confounding conclusions, for example if a different set of 'contaminants' were returned for a sample under comparison, how would the researcher know if these were new 'contaminants' or relocalisation of proteins to the nucleus?

We completely understand this reviewer's concern and hope that our additional analysis of quantification robustness and reproducibility of non-nuclear proteins in the previous responses addressed this point. As an enrichment method we cannot exclude a degree of contamination, but the high stability of non-nuclear annotated proteins in FrozONE is, at the least, similar with the other conventional methods we included in our benchmarking.

Furthermore, changes in protein abundance captured by the FrozONE method could be downstream of multiple different scenarios:

- i. Increase in abundance of a nuclear protein
- ii. Increase in abundance of a contaminant protein
- iii. Relocalisation of protein from contaminant compartment to nucleus with increase in overall abundance
- iv. Relocalisation of protein from contaminant compartment to nucleus with decrease in overall abundance

Conversely, there could be relocalisation of protein from 'contaminant' to nucleus and vice versa with no net change in abundance. It will be very difficult to determine which of the above scenarios is supported by the data acquired without better analysis of the proteome captured by the FrozONE method. Without an assessment of nuclear leakage, an assessment of the impact of nuclear processes by a specific perturbation or knowledge of differences between cell types and tissues will be incomplete.

We thank the reviewer for the elaborate description of scenarios that could lead to protein abundance changes in FrozONE, or any non-completed nuclear analysis strategy for that matter. We agree that case ii, iii and iv, all stemming from the possibility of enriching non-nuclear contaminant proteins, are a major limitation of these investigations that must be acknowledged when performing them. Still, our extensive analyses (shown

in the manuscript and the responses above) assessing stability and reproducibility of nuclear and non-nuclear annotated proteins, taking into consideration the incompleteness of those annotations, support the claim of our manuscript: the FrozONE method can reliably compete with conventional nucleus enrichment methods to provide robust, fast and scalable nuclear proteomic investigations.

Minor points:

1. There are no citations for the three centrifugation methods used by the authors with which to benchmark FrozONE

The missing references are now added.

2. Figure 1E - why were these 5 categories chosen? Mistake in labelling 'shuttling'

Our rationale for using such diverse categories was to show a representative group of proteins, often difficult to precisely quantify due to low expression, that play important nuclear functions in diverse healthy and pathological biological processes. We have now added this rationale to our manuscript (highlighted on page 4, "We then investigated ...").

3. Figure S1A - the coloring of the scale of Pearson Correlation scores is not distinct enough.

We thank the reviewer for pointing this out, we have now opted to indicate the Pearson Correlation values themselves.

4. The enrichment analysis presented in **Fig 2C** shows many processes that are unlikely to take place in the nucleus or in fact in the specific tissue in which it is highlighted. Can the authors comment on this observation?

We understand the point addressed here by the reviewer, this confusion is due to the nature of the protein annotations, as they are not tissue specific and protein networks can serve different functions in different tissues. The enrichment analysis could thus indicate functions that these protein networks serve in other tissues. For example, in the brain panel of **Fig 2C** the KEGG term gastric acid secretion is enriched even though it is obviously not a brain process. However, major components of this KEGG pathway are part of the Calcium signaling involving CAMK, Calmodulin, phosphatidylinositol phosphodiesterases, adenylate cyclases, guanine nucleotide binding proteins and PKC as well as ion transporter channels. All of these proteins are also integral to brain physiology²² can also be found in the nucleus²³⁻²⁷, and are therefore identified as brain markers in our dataset when compared to liver and kidney.

Regarding processes unlikely to take place in the nucleus, we found that most of the proteins from processes included in our enriched annotations contain nuclear localization. For example, if we repeat the enrichment analysis with proteins exclusively annotated as nuclear 6 out of the 7 brain terms still show statistically significant enrichment. Similarly, in the liver, although metabolic enzymes are not conventionally considered to be located in the nucleus, there are studies showing their nuclear localization, such as for Cytochrome P450^{14,28}.

Taking all this into consideration we believe our observations to be specific and valid. And our data is displayed with transparency showing unbiased results based on statistically significant cut-off thresholds.

We have added these aspects to the revised manuscript (highlighted on page 5, "Annotations of ...").

5. Figure 2D. Why were the papers (Sharma et al, 2015 Sharma et al, 2015 Azimifar et al, 2014; Sigdel et al, 2020) chosen and how did the authors define these markers as nuclear in their papers? The authors claim that the FrozONE performs well in comparison with these studies in terms of coverage, but two of them are 9-10 years old and mass spectrometry sensitivity has moved on a long way since these papers were written.

We thank the reviewer for raising this concern. These papers were selected because they profiled individual cell populations in the exact tissues we used for our benchmarking tests. Despite being published some years ago, they employed extensive sample fractionation to produce a great depth of protein quantification that standard current single shot methods do not regularly reach.

Regarding nuclear localization, as the original authors have not investigated subcellular localization, the matched datasets we filtered for proteins annotated as nuclear according to the Uniprot Keywords and COMPARTMENTS database²⁹ as consistently used in our analyses.

6. Page 6, 3rd paragraph. The authors claim that FrozONE 'ensures unbiased enrichment of nuclei from different cell types'. The data as represented do not support this claim as the contribution of cell type to each tissue sample is not corrected for, and also as stated above, nuclear leakage and incomplete capture of the nuclear proteome plus differential levels of contamination could result in biased results.

We agree with the reviewer that this comment was an overestimation. We have accordingly modified our statements in the revised manuscript. We have also included an additional paragraph assessing cell type distribution within tissue using our quantitative data in a highlighted paragraph in page 5 ("In addition ...").

7. In the comparison of the High Fat Diet (HFD) versus the Control Diet (CD), it would have been good to look at the total change in protein abundance not just what was retrieved by FrozONE. As stated above, it could be that the perceived change in a nuclear protein could simply be to do with its relocalisation rather than its absolute amount. Once again, it is also important for the authors to show the terms enriched for the non-nuclear 'contaminants'. Do these proteins also reflect metabolic re-wiring promoted by the HFD?

We indeed agree on the importance of assessing the enrichment of any potential contaminant that FrozONE captures which can lead to overestimation of the "nuclear" metabolic effects. To answer this matter, we have taken the significantly upregulated non-nuclear proteins in both diets and performed the Fisher exact test for annotation enrichment against all non-nuclear proteins present in the comparison (background). As can be appreciated from **Figure R2C8** below, none of the processes that we observed with nuclear proteins (**Table S4**), such as Lipid metabolism, Lipid- and Fatty acid biosynthesis, can be found when using non-nuclear proteins for the enrichment, suggesting that the non-nuclear contaminant fraction of FrozONE does not reflect the metabolic rewiring induced by the NASH diet and that these indeed constitute nuclear events.

Enriched terms in Control Diet (Non-Nuclear) [Uniprot's Keywords]

Enriched terms in NASH diet (Non-Nuclear) [Uniprot's Keywords]

Figure R2C8. Barplot representing enriched Uniprot's Keywords from significantly upregulated non-nuclear proteins in each diet (control diet left, high fat diet right). Results were filtered for at least 5 proteins in each category and an enrichment factor bigger than 1 (FDR = 0.05).

8. In the methods section, to digest proteins to peptides were trypsin and Lys-C endoproteases added simultaneously or consecutively? 🍌

Yes, both enzymes were added simultaneously to digest samples during overnight incubation, this is now specified in the revised manuscript.

9. What is SDP-RPS? 🍌

SDP-RPS stands for Styrenedivinylbenzene-Reversed Phase Sulfonate. It is a solid-phase extraction material commonly used for peptide desalting and cleaning during proteome sample preparation. We have now referenced the publication containing an in-depth explanation of this stage tipping procedure³⁰ in the method section of the revised version of this manuscript.

10. For missing value imputation, what was the rationale behind using the smallest intensity value found in this merged dataset, rather than, for example K-NN based approach? 🍌

We only used the smallest value imputation method for proteins completely absent in the control diet included in the heatmap of **Fig 4**. That was a purely visual strategy so that after z-scoring the detected values in the high fat diet would fit the scale and the imputed values were then removed again after z-scoring. For our imputations upstream of statistical testing, we instead used an imputation strategy that places in each missing value of a sample random values from a defined window of the lowest intensity values from the normal distribution of this sample, simulating a region of values at the border of detection limit.

11. Could the authors provide a cost benefit analysis of FrozONE against the other methods including, sample amount, cost of kits etc.. and time. 🍌

Certainly, concerning only required material (excluding tabletop centrifuges, common consumables, proteases inhibitors, etc) the 388€ NUC101-1KT for 40 samples brings FrozONE to an overhead cost of 9.7€/sample while the cost of the buffers for the sucrose gradient is ~2.21€/sample (see table below). However, since the sucrose method requires an ultracentrifuge, this adds additional instrumentation cost (current price of the Beckman Optima XPN 80 with SW40Ti Rotor is ~ 53 000€). The experimental time required for FrozONE's centrifugation steps is roughly 1h, shorter than the sucrose density centrifugation. However, the main

advantage is that conventional tabletop centrifuges can usually hold a minimum of 24 samples, while UZ rotors for the required volumes only hold 6 tubes, making the UZ methods up to at least 4x less scalable. Finally, although we did not systematically test sample amount requirements and outputs, we generally observed that from a similar amount of tissue (~300mg) the sucrose method yielded ~100µg and FrozONE reached ~400µg of total nuclear protein.

Sucrose buffer:	Substance	Cushion buffer (2.05M)	Homogenization buffer (2.2M)	list prize	prize/100ml	ref
	Sucrose	70.1g	75.3g	97.8€/500g	13.69	https://www.sigmaaldrich.com/DE/de/product/sigma/s0389
	1M HEPES pH 7.6	1 mL	1 mL	46.6€/10g	1.11	https://www.sigmaaldrich.com/DE/de/product/sigma/54457
	3M KCl	500 µl	500 µl	70.8€/100g	0.08	https://www.sigmaaldrich.com/DE/de/product/sigma/p9541
	0.5M EDTA	400 µl	400 µl	35.5€/100ml 500mM	0.14	https://www.sigmaaldrich.com/DE/de/product/mm/324504
	0.1M Spermine	150 µl	150 µl	184€/5g	0.11	https://www.sigmaaldrich.com/DE/de/product/sigma/85590
	6.4M Spermidine	50 µl	50 µl	46.3€/g	2.15	https://www.sigmaaldrich.com/DE/de/product/sigma/s2626
	Water	Fill to 100 mL	Fill to 100 mL		17.29	
					13ml/sample = 2.21€	
FrozONE:	5mL of 388€/200ml NUC101-1KT 388€ (https://www.sigmaaldrich.com/DE/de/product/sigma/nuc101) = 9.7€/sample					

12. The annotation of supplementary tables and data analysis approaches could be improved upon, for example, what do the values in Table 1 correspond to? I assume raw intensity values, without any normalization? The details of normalization, and filtering could be a little clearer overall. 🙌

We thank the reviewer for this comment and agree that we missed to precisely describe this relevant information. We have now included a detailed description of the content of each tab for all supplementary tables.

Reviewer #3:

Direct processing of large amount of fresh tissue material is frequently perceived as an absolute necessity for a robust proteomic profiling of tissues. This becomes even more complicated when extracting proteins from the nuclei or other cellular compartments. To extract nuclear proteins, sample processing also involves laborious and time-consuming ultracentrifugation step to extract tissues protein. Here, Huschet, L.A. and Kliem, F.P. et al. developed a workflow for proteomic analysis for frozen tissues, named FrozONE, by combining mechanical nuclei extraction using commercial Qiagen TissueLyser II/Sigma Nuclei EZ Prep and MS-based quantitative proteomic with data independent analysis (DIA). Using this approach, the authors showed comparably robust nuclear proteomic analysis between fresh and frozen tissues, extracted either by the ultracentrifugation step or by the commercial kit from different tissue origins. Finally, the authors applied their workflow to investigate changes in the liver nuclear proteome in response to high fat diet and from different brain regions. This is a very solid manuscript with data of high quality and the workflow is surely useful for the community; however I feel that both the methodological or biological advance is not sufficient for publication in Mol Systems Biology. Apart from the simplicity FrozONE extraction using TissueLyser II/ Nuclei EZ Prep kit, the proposed workflow lacks of significant novelty and superiority over existing technique. Furthermore, the authors appear to push the superiority of their workflow by not highlighting the increase sensitivity of DIA in comparison to DDA approach in their MS analysis.

We greatly appreciate this reviewer's thoughtful and detailed assessment of our manuscript. To answer the last raised point, we would like to clarify that the objective of our manuscript was not to highlight the increased sensitivity of the DIA method over the DDA. We hope we made clear that DIA measurements were performed in all cases for all methods to thus merely benchmarked the performance of the nuclear enrichment methods among each other using the same MS strategy. We do agree that FrozONE does not display an immense superiority over the other methods when purely judging the quantification depth and nuclear fraction enrichment. However, our extensive comparisons and analyses showed that the superiority of FrozONE over other methods lay on the great performance with frozen material, speed and, very importantly, the scalability potential. All this together makes FrozONE the only compatible nuclei enrichment method to apply to large

scale nuclear proteomics studies of any type of frozen specimens and biopsies. We therefore believe that our method would be of great potential to investigate aberrant protein nuclear changes that are the basis of diverse pathological conditions of high prevalence such as cancer, metabolic disorders, ALS, FTD and Huntington's and Alzheimer's diseases.

As a methods paper, the "Methods and Protocols" section of the manuscript is written poorly and lacks the necessary detail. For instance, the nuclei enrichment section failed to mention the starting amount of tissue material for all the enrichment protocol (i.e. sucrose gradient vs iodixanol gradient vs FrozONE). What is Nuclei EZ Prep buffer used in the FrozONE section? Is it the Nuclei EZ "lysis" or the "storage" buffer used for the extraction? Buffer compositions is poorly describe in the "Proteome samples preparation"

We thank the reviewer for this comment and we now make sure to precisely describe FrozONE's method in a protocol-like style providing additional key explanations in the section "Proteome samples preparation".

The paper should compare cost and time benefit between FrozONE and existing sucrose gradient extraction.

Certainly, concerning only material that one method and not the other needs (excluding tabletop centrifuges, common consumables, proteases inhibitors, etc) the 388€ NUC101-1KT for 40 samples brings FrozONE to an overhead cost of 9.7€/sample while the cost of the buffers for the sucrose gradient is ~2.21€/sample (see table below). However, since the sucrose method requires an ultracentrifuge, this major investment needs to be made first (currently Beckman Optima XPN 80 with SW40Ti Rotor ~ 53 000€). The experimental time required for FrozONE's centrifugation steps is roughly 1h, shorter than the sucrose density centrifugation. However the main advantage is that conventional tabletop centrifuges can usually hold at least 24 samples, while UZ rotors for the required volumes only hold 6 tubes, making the UZ methods up to at least 4x less scalable. Finally, sample amount requirements and outputs have not been systematically tested, but sucrose usually gave ~100µg and FrozONE ~400µg total nuclei protein for a ~300mg tissue (brain).

Sucrose buffer:	Substance	Cushion buffer (2.05M)	Homogenization buffer (2.2M)	list prize	prize/100ml	ref
	Sucrose	70.1g	75.3g	97.8€/500g	13.69	https://www.sigmaaldrich.com/DE/de/product/sigma/s0389
	1M HEPES pH 7.6	1 mL	1 mL	46.6€/10g	1.11	https://www.sigmaaldrich.com/DE/de/product/sigma/54457
	3M KCl	500 µl	500 µl	70.8€/100g	0.08	https://www.sigmaaldrich.com/DE/de/product/sigma/p9541
	0.5M EDTA	400 µl	400 µl	35.5€/100ml 1 500mM	0.14	https://www.sigmaaldrich.com/DE/de/product/mm/324504
	0.1M Spermine	150 µl	150 µl	184€/5g	0.11	https://www.sigmaaldrich.com/DE/de/product/sigma/85590
	6.4M Spermidine	50 µl	50 µl	46.3€/g	2.15	https://www.sigmaaldrich.com/DE/de/product/sigma/s2626
	Water	Fill to 100 mL	Fill to 100 mL		17.29	
					13ml/sample = 2.21€	
FrozONE:	5mL of 388€/200ml NUC101-1KT 388€ (https://www.sigmaaldrich.com/DE/de/product/sigma/nuc101) = 9.7€/sample					

References:

1. Goldman, L. & Schafer, A. *Goldman's Cecil Medicine*. (2012).
2. Balzer, M. S., Rohacs, T. & Susztak, K. How Many Cell Types Are in the Kidney and What Do They Do? *Annu. Rev. Physiol.* **84**, 507–531 (2022).
3. Ochocka, N. & Kaminska, B. Microglia Diversity in Healthy and Diseased Brain: Insights from Single-Cell Omics. *Int. J. Mol. Sci.* **22**, 3027 (2021).
4. Sun, W. *et al.* SOX9 Is an Astrocyte-Specific Nuclear Marker in the Adult Brain Outside the Neurogenic Regions. *J. Neurosci.* **37**, 4493–4507 (2017).
5. Valério-Gomes, B., Guimarães, D. M., Szczupak, D. & Lent, R. The Absolute Number of Oligodendrocytes in the Adult Mouse Brain. *Front. Neuroanat.* **12**, 90 (2018).
6. MacParland, S. A. *et al.* Single cell RNA sequencing of human liver reveals distinct intrahepatic macrophage populations. *Nat. Commun.* **9**, 4383 (2018).
7. Sitia, G. *et al.* Kupffer Cells Hasten Resolution of Liver Immunopathology in Mouse Models of Viral Hepatitis. *PLOS Pathog.* **7**, e1002061 (2011).
8. Bogdanos, D. P., Gao, B. & Gershwin, M. E. Liver Immunology. *Compr. Physiol.* **3**, 567–598 (2013).
9. Du, W. & Wang, L. The Crosstalk Between Liver Sinusoidal Endothelial Cells and Hepatic Microenvironment in NASH Related Liver Fibrosis. *Front. Immunol.* **13**, 936196 (2022).
10. Giampieri, M. P., Jezequel, A. M. & Orlandi, F. The lipocytes in normal human liver. A quantitative study. *Digestion* **22**, 165–169 (1981).
11. Li, X., Wang, Z. & Klaunig, J. E. Modulation of xenobiotic nuclear receptors in high-fat diet induced non-alcoholic fatty liver disease. *Toxicology* **410**, 199–213 (2018).
12. Ghoneim, R. H., Ngo Sock, E. T., Lavoie, J.-M. & Piquette-Miller, M. Effect of a high-fat diet on the hepatic expression of nuclear receptors and their target genes: relevance to drug disposition. *Br. J. Nutr.* **113**, 507–516 (2015).
13. Sinha, R. A. Targeting nuclear receptors for NASH/MASH: From bench to bedside. *Liver Res.* **8**, 34–45 (2024).
14. Wang, J. *et al.* Nuclear Proteomics Uncovers Diurnal Regulatory Landscapes in Mouse Liver. *Cell Metab.* **25**, 102–117 (2017).
15. Go, C. D. *et al.* A proximity-dependent biotinylation map of a human cell. *Nature* **595**, 120–124 (2021).
16. Thul, P. J. *et al.* A subcellular map of the human proteome. *Science* **356**, eaal3321 (2017).
17. Cho, N. H. *et al.* OpenCell: proteome-scale endogenous tagging enables the cartography of human cellular organization. *Science* **375**, eabi6983 (2022).
18. Christoforou, A. *et al.* A draft map of the mouse pluripotent stem cell spatial proteome. *Nat. Commun.* **7**, 9992 (2016).
19. Erdmann, G., Schütz, G. & Berger, S. Inducible gene inactivation in neurons of the adult mouse forebrain. *BMC Neurosci.* **8**, 63 (2007).
20. Mo, A. *et al.* Epigenomic Signatures of Neuronal Diversity in the Mammalian Brain. *Neuron* **86**, 1369–1384 (2015).
21. Sharma, K. *et al.* Cell type- and brain region-resolved mouse brain proteome. *Nat. Neurosci.* **18**, 1819–1831 (2015).
22. Kawamoto, E. M., Vivar, C. & Camandola, S. Physiology and Pathology of Calcium Signaling in the Brain. *Front. Pharmacol.* **3**, (2012).
23. Ma, H. *et al.* γ CaMKII shuttles Ca²⁺/CaM to the nucleus to trigger CREB phosphorylation and gene expression. *Cell* **159**, 281–294 (2014).
24. Stallings, J. D., Tall, E. G., Pentyala, S. & Rebecchi, M. J. Nuclear Translocation of Phospholipase C- δ 1 Is Linked to the Cell Cycle and Nuclear Phosphatidylinositol 4,5-Bisphosphate*. *J. Biol. Chem.* **280**, 22060–22069 (2005).
25. Parkinson, N. A. & Bolsover, S. R. A nuclear location for Ca²⁺-activated adenylyl cyclases I and III in neurones. *Brain Res. Mol. Brain Res.* **91**, 43–49 (2001).
26. Sample, V. *et al.* Regulation of Nuclear PKA revealed by spatiotemporal manipulation of cAMP. *Nat. Chem. Biol.* **8**, 375–382 (2012).
27. Matzke, A. J. M., Weiger, T. M. & Matzke, M. Ion Channels at the Nucleus: Electrophysiology Meets the Genome. *Mol. Plant* **3**, 642–652 (2010).
28. Bresnick, E., Boraker, D., Hassuk, B., Levin, W. & Thomas, P. E. Intranuclear Localization of Hepatic Cytochrome P448 by an Immunochemical Method. *Mol. Pharmacol.* **16**, 324–331 (1979).
29. Binder, J. X. *et al.* COMPARTMENTS: unification and visualization of protein subcellular localization evidence. *Database* **2014**, bau012 (2014).
30. Rappsilber, J., Mann, M. & Ishihama, Y. Protocol for micro-purification, enrichment, pre-fractionation and storage of peptides for proteomics using StageTips. *Nat. Protoc.* **2**, 1896–1906 (2007).

November 11, 2024

Re: Life Science Alliance manuscript #LSA-2024-03130-T

Prof Maria S. Robles
Institute of Medical Psychology, Faculty of Medicine, LMU Munich
Germany

Dear Dr. Robles,

Thank you for submitting your manuscript entitled "FrozONE: Quick Cell Nucleus Enrichment for Comprehensive Proteomic Analysis of Frozen Tissues" to Life Science Alliance. We invite you to submit a revised manuscript addressing the following Reviewer comments:

- Address Reviewer 2's comments, except for the concern related to the kit used.
- Address Reviewer 3's comments, except for comments #2 & 3.

Thank you for this interesting contribution to Life Science Alliance. We are looking forward to receiving your revised manuscript.

Sincerely,

- A letter addressing the reviewers' comments point by point.
- An editable version of the final text (.DOC or .DOCX) is needed for copyediting (no PDFs).
- High-resolution figure, supplementary figure and video files uploaded as individual files: See our detailed guidelines for preparing your production-ready images, <https://www.life-science-alliance.org/authors>
- Summary blurb (enter in submission system): A short text summarizing in a single sentence the study (max. 200 characters including spaces). This text is used in conjunction with the titles of papers, hence should be informative and complementary to the title and running title. It should describe the context and significance of the findings for a general readership; it should be written in the present tense and refer to the work in the third person. Author names should not be mentioned.
- By submitting a revision, you attest that you are aware of our payment policies found here: <https://www.life-science-alliance.org/copyright-license-fee>

B. MANUSCRIPT ORGANIZATION AND FORMATTING:

Point by point reply to reviewers' comments
#LSA-2024-03130-T

Reviewer #2:

1- I do not accept the point about the authors make about their kit being no different to several other kits readily employed by the proteomics community. The TMT and SILAC kits are not comparable. The contents of the TMT tag kits and also the amino acid sold to SILAC label proteins are much simpler than the kit used in the FrozONE method.

2- Figure R2C1 is confusing. A Venn diagram would show what is in common between the two methods.

Following the reviewers' suggestion we have now depicted in venn diagrams the comparisons between our method and the standard sucrose fresh contained in figure R2C1/S2A. This new representation method also highlights the large overlap between nuclear proteins quantified with the two different methods. We consider that even though Venn diagrams are easier to understand due to their familiarity, the representation in this way is more complex and will require larger space in the figure. Therefore, we decided to keep the original upset plot in FigS2A, with the possibility of exchanging it for the venn diagrams if this review prefers.

Figure R2.1 Venn diagrams showing the overlap in protein identities between FrozONE and Sucrose fresh quantifications in Brain, Liver and Kidney, subdivided into all proteins in the top, nuclear annotated proteins in the middle and not nuclear annotated in the bottom row.

3- There is a substantial proportion of proteins that are not nuclear. I find figure R2C1 very confusing.

Indeed, from the total number of proteins quantified with all nuclear enrichment methods a proportion of them cannot be classified as nuclear using publically available protein annotations. As we discussed in the previous revision of this manuscript, the proportion of not nuclear annotated proteins can be attributed to two potential sources: a) Nuclear proteins that are not annotated as nuclear due to uncompleted annotations. Our data from the efficient nuclei enrichment method, FANS, hint in that direction (see below Fig. R2.4).

b) Proteins from other subcellular compartments that are present in the enriched nuclear fractions. This is indeed possible, as the reviewer pointed out, as all the methods we have employed, including FrozONE, are not producing completely isolated nuclei but rather nuclear enriched fractions (see new Fig. S3). Below, in response to this reviewers' concerns we have provided data showing that some non-nuclear annotated proteins can be classified as part of other cellular organelles, however the overall fraction is not very substantial. Moreover, and very important for this benchmarking study, the proportion of proteins annotated from other organelles in FrozONE samples is lower or equal to those found with the standard sucrose fresh method (new Fig.S3) as well as the more specific enrichment/isolation method based on sorting (Fig. R2.4).

4- The authors have done a lot of analysis to show that the contaminants are reproducible, but without showing where they come from in the cell. My comments about the appropriateness of the method still stand - this is not a method that gives high quality nuclear preparations, where approx. a third of the proteins identified are likely to be not nuclear.

We acknowledge that we have previously not specifically represent whether non-nuclear annotated proteins (which could be potential contaminants or nuclear proteins with non existing nuclear annotations to date, see our answer above) can be classified from other cellular compartments, information that we now provide and included in a new figure (R2.2 below, new Fig. S3) in the revised version of the manuscript. Using the same database (COMPARTMENTS) we have now highlighted nuclear annotated proteins and proteins not annotated as nuclear but in other cellular organelles, such as those i) physically associated with the nucleus and common isolation confounder endoplasmic reticulum (PMID: 33918601, PMID: 38767195), ii) in close proximity such as Golgi apparatus (PMID: 21071196) and iii) cytosolic organelles known to tether to (peri-)nuclear structures such as mitochondria (PMID: 33355129) and lysosomes (PMID: 32624271). As shown below, the total number of proteins annotated as localizing to these organelles is not substantial and similar when comparing FrozONE and Sucrose fresh (Fig. S3A). Furthermore, when directly assessing the nuclei enrichment capability of both methods by comparing protein intensities in nuclear fractions versus whole tissue lysates one can clearly observe that FrozONE enriches nuclear proteins equal or better, depending on the tissue, than sucrose fresh (Fig. S3B). In contrast, proteins annotated from other cellular organelles show a clear de-enrichment in FrozONE nuclear preparations, again to a better or similar degree as sucrose fresh preparations (Fig. S3C). We can thus conclude that no organelle is prone to substantially "contaminate" FrozONE preparations to a higher degree than in current standard nucleus enrichment methods.

To make it clear to this review, the aim of establishing FrozONE was not to produce highly isolated nuclear preparations but rather enrich them to the same or better degree that other standard widely used methods, specifically from frozen tissues. We hope that our data can now show that FrozONE provides the same or better degree of nuclei enrichment from frozen tissues compared to fresh material with the great advantage of being scalable, faster and more economical.

Figure R2.2.A. Barplot of total number of proteins quantified within biological replicates ($n = 3$) in FrozONE or sucrose gradient with fresh tissue, annotated based on different subcellular compartments (COMPARTMENTS database). For non-nuclear compartments only proteins without the annotation “Nucleus” were used. B & C. Volcano plots comparing protein intensities between FrozONE or Sucrose fresh nuclear preparations and whole cell lysates (WCL) for each tissue. The fill colors marked proteins with the same annotations as shown in A. (This Figure is now a new FigS3 in the revised manuscript and its description included in page 4 of the revised version of the manuscript.)

5- Figure R2C4 is also very confusing. In the bottom lefthand barplot they seem to have a done, to my min a back to front analysis. They have used another data set (Wang et al) and shown what proteins in that dataset which are part of other compartments, are not enriched in their data, **rather than what compartments their contaminants are from.**

We acknowledge that this back to front comparison was not the most optimal approach to answer the reviewer's concerns. We have now re-analysed the comparison with a "front analysis" considering nuclear and non-nuclear organelle annotations in proteins present in our dataset but not in the Wang et al dataset as the reviewer suggested. This comparison supports our overall findings as we observed that these proteins are preferentially annotated as nuclear rather than other indicated organelles (see figure R2.3).

Figure R2.3 Barplot of number of proteins exclusively quantified within biological replicates ($n = 3$) in FrozONE or sucrose gradient with fresh tissue from livers, but missing in Wang et al dataset, annotated with different subcellular compartments (COMPARTMENTS database). For non-nuclear compartments only proteins without the annotation "Nucleus" were used.

6- The comparison with the Go et al data is also confusing - again back to front. They show which proteins overlap, but not where the protein in the FrozOnNE data set do not overlap. Instead, they look at potential contamination in the Go et al data rather than characterize the non nuclear proteins in their own data.

We apologize if our comparison was not sufficiently clear. However, based on this review's original comment "I would like to have seen overlay of the data with other [...] data sets [...]. Without these data, the potential users of the method would have no idea what portion of the nuclear is being captured by this method..." we centered our analysis in showing the portion of the "nuclear proteome" defined by the Go et al data that is captured or not in FrozONE samples. We answered this point directly to the reviewer but also added this comparison in our manuscript, in a figure and table (Fig. S7, Table S1). Making nevertheless clear that we think this type of comparison is not optimal as it correlates our mouse tissue nuclei proteomes to Go et al. human HEK293 protein classification.

We hope that our new analysis, shown now in new Fig. S3 (see R2.2 above) can provide a clear answer to this new comment of the reviewer.

7- The comparisons with the other datasets listed are misleading with variable overlap of identified nuclear proteins, but no information about the **other compartments** that the FrozONE identified proteins map to.

We agree with the reviewer, that making sure that no other organelles contaminate our FrozONE preparation was crucial and hope that the data of our analyses, now included in the new Fig. S3 clarified this concern. As briefly mentioned above, the variable overlap we observed between the published studies and our data could be due to different parameters: i) the diverse cell/tissue sources being used, ii) and/or the different applied methodologies iii) and/or the respective confounders associated with each study.

8- The authors go on to compare their data with that generated from florescent activated nucleus sorting. The overlap is good, but does not allay my concerns about the contamination issues. **What are the 2094 proteins** that do not overlap in figure R2C7?

This is indeed an interesting question for which we did not provide information. We have now reevaluated this data and show how these 2094 proteins are distributed across major organelles using the COMPARTMENTS database. We found that, despite not being contained in FANS proteomes, almost 50% of these FrozONE exclusive proteins have nuclear annotation (Fig. R2.4), similar to the overall percentage seen in the rest of FrozONE, sucrose fresh and even FANS nuclei exclusive or total fractions (Fig. R2.5). Data in agreement with the overall nuclei enrichment, and depletion of other organelles we observed when compared to whole cell lysates (new Fig. S3). Overall we can conclude that the FrozONE proteins not contained in FANS nuclei are not due to substantial contamination from other non-nuclear organelles but rather belong to nuclear proteins from other non-neuronal brain cell types as in fact revealed by the enrichment analysis using cell type markers (Fig. R2.4 and Fig. S5 in manuscript).

Figure R2.4 Overlap of proteins quantified in FrozONE nuclear preparations from hippocampus and with nuclei from hippocampus sorted by fluorescence activated nucleus sorting (FANS) using SUN1-GFP. Top right: Results of the enrichment analysis (using cell type markers from Sharma et al, 2015) of exclusive FrozONE quantified proteins. Bottom right: Barplot showing the number of exclusive FrozONE quantified proteins according to their COMPARTMENTS database annotation. For non-nuclear compartments only proteins without the annotation "Nucleus" were used.

Furthermore, if we do the same comparison with the FANS exclusive proteins (971), we observe a very similar distribution of proteins annotated in those organelles (e.g. ~10% Mitochondrion) as in FrozONE exclusive proteins, mirroring the distribution of the overall FANS proteins (Fig. R2.5). This indicates that i) proteins that do not overlap among FrozONE and FANS samples are not overrepresented in non-nuclear compartments and ii) that FrozONE does not produce less enriched nuclear proteomes compared to a more efficient enrichment method as FANS.

Figure R2.5 Barplots showing total number (left; total 3864) and exclusive (right; total 971) FANS quantified proteins compared to FroZONE corresponding to their annotation to different subcellular compartments (COMPARTMENTS database). For non-nuclear compartments only proteins without the annotation “Nucleus” were used.

9- In short, the authors have not shown **where their non-nuclear annotations come from**. It is one thing to show partial overlap of the nuclear proteins identified, (which in many cases is less than two thirds) but my point was about the contaminants. Let's say I use this method to show changes in the nuclear proteome upon drug treatment. It may be the change I observe is within the contaminating fraction - I would not be able to distinguish relocalization to a contaminating fraction over the nuclear fraction of the sample.

We hope with the new analysis we answer this general comment of this reviewer. Overall our data showed that FroZONE is able to efficiently enrich nuclear proteins, equal or better than the gold classical method. And at the same time, it depleted, in most of the cases to a better degree than the standard density gradient method, proteins from organelles potential co-founders in the majority of nuclei enrichment methods.

10- The authors do not adequately address my concerns at the bottom of page 9 of their rebuttal document.

We appreciate the reviewer’s concerns regarding the potential contribution of confounder proteins in the data. Since FroZONE, as sucrose gradient, is an enrichment and not an isolation method, we acknowledge that some confounders, based on missed annotated proteins and/or contaminants, cannot be completely ruled out. This limitation is inherent to any nuclear enrichment method, including the widely used sucrose gradient. Still, our new comparative analysis of nuclear fractions with whole cell lysates (new Fig. S3B & S3C) show that most proteins from other organelles are, to a very good degree, de-enriched in FroZONE samples. One way to distinguish between specific nuclear changes from those due to abundance or localization changes in potential cofounder/contaminants will be to additionally assay whole cell lysates alongside nuclear-enriched fractions in experimental setups. This dual approach would allow researchers to disentangle changes in nuclear proteomes from broader cellular protein dynamics, providing more robust conclusions.

We have now included this suggestion in the updated manuscript as follows on page 10 & 11:

“Given the inherent nature of any nuclear enrichment, it cannot be expected to obtain a grade of purity that resembles the one of an isolation method. Despite FrozONE’s efficiency in depleting proteins from non-nuclear organelles (Fig. S3C) in similar or better degree than the gold standard method of sucrose gradient, it cannot fully segregate between different possibilities, such as changes in the levels of actual non-nuclear proteins due to abundance change within or re-localization to or from a contaminating organelle. This confounding factor could be further addressed by complementing the quantification of FrozONE-based nuclear enriched fractions with whole cell lysates to better distinguish nuclear-specific from confounder changes.

11- In short, the authors have created a subcellular enrichment method that is reproducible. It uses a commercial kit, and it is not clear therefore how the enrichment works and thus is very reliant on the availability and provenance of the kits. The method seems to partially enrich the nuclear proteome, but also a substantial proportion of contaminating subcellular compartments. Its use in characterizing the nuclear proteome upon perturbation is thus limited.

We have now addressed the concerns of the reviewer by analysing organelles that are part of or tethering to the (peri-)nuclear complex (PMID: 33918601, PMID: 38767195, PMID: 21071196, PMID: 33355129, PMID: 32624271) that could be present in nuclear enriched fractions. Here, we have shown that our method yields a similar degree of other-organelle annotated proteins and depletion of them when compared to whole cell lysates, that the gold standard and a more efficient, FANS, method of nuclear enrichment. Hence showcasing that FrozONE is in any way more biased for contaminations of any specific cytoplasmic organelle.

As an enrichment method, we cannot claim, and we did not so far, that FrozONE produces pure nuclear fractions. However, its efficiency in nuclear enrichment capacity, similar to the gold standard, can allow researchers to obtain qualitative nuclei preparations comparable to other classical methods with the advantage of using frozen material, being faster and scalable.

Reviewer

#3:

The re-submitted version of the manuscript is improved, the data is of high quality and analysis performed to a high standard and showed in clear figures. However, the technological and biological advancement that this manuscript describes in my opinion is not of enough novelty to be published in this journal.

- To further improve the manuscript, in Figure 1 the authors should take annotations from Protein Atlas for cellular localization and show the distribution of proteins identified in FrozONE and comparative tissue nuclei isolation methods to **all compartments**. Currently, Figure 1B only shows nuclear vs non-nuclear annotation. It would be

important to further break non-nuclear to different annotations to see whether any of the methods had a particular bias to another compartments for example mitochondria.

We have now revisited our data and plotted it in a slightly different way to respond to this (as well as a comment of the second reviewer). Using the same database (COMPARTMENTS) we have now highlighted nuclear annotated proteins and protein uniquely annotated in other cellular organelles, such as those i) physically associated with the nucleus as the endoplasmic reticulum, ii) in close proximity as Golgi and iii) frequent co-founders in nuclear fractions, mitochondria and lysosomes. As shown below in Fig R3.1 (new Fig.S3 in the revised manuscript), the total number of proteins annotated as localizing to these organelles is not very substantial and very similar when comparing FrozONE and Sucrose fresh methods (Fig. S3A). Furthermore, when directly assessing the nuclei enrichment capability of both methods by comparing protein intensities in nuclear fractions versus whole tissue lysates one can clearly observe that FrozONE enriches nuclear proteins equal or better, depending on the tissue, than sucrose fresh (Fig. S3B). In contrast, proteins annotated from other cellular organelles show a clear depletion in FrozONE nuclear preparations, again to a better or similar degree as sucrose fresh preparations (Fig. S3C). We can thus conclude that no organelle is prone to substantially “contaminate” FrozONE preparations to a higher degree than in the current standard nucleus enrichment method.

Figure R3.1 A. Barplot of total number of proteins quantified within biological replicates ($n = 3$) in FrozONE or sucrose gradient with fresh tissue, annotated based on different subcellular compartments (COMPARTMENTS database). For non-nuclear compartments only proteins without the annotation “Nucleus” were used. B & C. Volcano plots comparing protein intensities between FrozONE or Sucrose fresh nuclear preparations and whole cell lysates (WCL) for each tissue. The fill colors marked proteins with the same annotations as shown in A. (This Figure is now a new FigS3 in the revised manuscript and its description included in page 4 of the revised version of the manuscript.)

- In addition, Protein Atlas annotations can be also used to look at different sub-nuclear compartments to look whether there is a bias for detection of specific nuclear sub-compartments.
- It would be also important to look at chromatin-associated protein recovery with FrozONE; which could for example be done using the data in EMBO J. 2014 Feb 17;33(6):648-664. doi: 10.1002/emboj.201387614 (Kustatscher et al).

- It is not necessary to state in the introduction that "Nuclei are one of the most important organelles in the cells.."

Following the reviewer's suggestion we removed this statement.

- In the results, it is stated that FrozONE performs remarkably compared to previous studies, but here it has to be acknowledged that DIA is used and that this is also a result of advancement in MS technology.

This is indeed a fair point about our comparisons with previous studies that we consider should be acknowledged. We have now mentioned this aspect in the main text.

- S1B has formatting problems.

We have generated a new version of this figure to correct the formatting problems.

December 2, 2024

RE: Life Science Alliance Manuscript #LSA-2024-03130-TR

Prof. Maria S. Robles
Ludwig-Maximilians-Universität München
Institute of Medical Psychology and BMC
Biomedical Center, Faculty of Medicine
Munich 80336
Germany

Dear Dr. Robles,

Thank you for submitting your revised manuscript entitled "FrozONE: Quick Cell Nucleus Enrichment for Comprehensive Proteomic Analysis of Frozen Tissues". We would be happy to publish your paper in Life Science Alliance pending final revisions necessary to meet our formatting guidelines.

- please be sure that the authorship listing and order is correct
- please upload your manuscript text as an editable doc file
- please add the Twitter handle of your host institute/organization as well as your own or/and one of the authors in our system
- please make the PRIDE dataset publicly accessible at this time, removing the need for the Reviewer access information in the Data Availability statement

A. FINAL FILES:

B. MANUSCRIPT ORGANIZATION AND FORMATTING:

Sincerely,

December 3, 2024

RE: Life Science Alliance Manuscript #LSA-2024-03130-TRR

Prof. Maria S. Robles
Ludwig-Maximilians-Universität München
Institute of Medical Psychology and BMC
Biomedical Center, Faculty of Medicine
Munich 80336
Germany

Dear Dr. Robles,

Thank you for submitting your Methods entitled "FrozONE: Quick Cell Nucleus Enrichment for Comprehensive Proteomic Analysis of Frozen Tissues". It is a pleasure to let you know that your manuscript is now accepted for publication in Life Science Alliance. Congratulations on this interesting work.

DISTRIBUTION OF MATERIALS:

Again, congratulations on a very nice paper. I hope you found the review process to be constructive and are pleased with how the manuscript was handled editorially. We look forward to future exciting submissions from your lab.

Sincerely,
